# META-LEARNING BAYESIAN NEURAL NETWORK PRIORS BASED ON PAC-BAYESIAN THEORY

## ABSTRACT

Bayesian deep learning is a promising approach towards improved uncertainty quantification and sample efficiency. Due to their complex parameter space, choosing informative priors for Bayesian Neural Networks (BNNs) is challenging. Thus, often a naive, zero-centered Gaussian is used, resulting both in bad generalization and poor uncertainty estimates when training data is scarce. In contrast, *meta-learning* aims to extract such prior knowledge from a set of related learning tasks. We propose a principled and scalable algorithm for meta-learning BNN priors based on PAC-Bayesian bounds. Whereas previous approaches require optimizing the prior and multiple variational posteriors in an interdependent manner, our method does not rely on difficult nested optimization problems, and moreover, it is agnostic to the variational inference method in use. Our experiments show that the proposed method is not only computationally more efficient but also yields better predictions and uncertainty estimates when compared to previous meta-learning methods and BNNs with standard priors.

## 1 INTRODUCTION

Bayesian Neural Networks (BNNs) offer a probabilistic interpretation of deep learning by inferring distributions over the model's weights (Neal, 1996). With the potential of combining the scalability and performance of neural networks (NNs) with a framework for uncertainty quantification, BNNs have lately received increased attention (Blundell et al., 2015; Gal & Ghahramani, 2016). In particular, their ability to express epistemic uncertainty makes them highly relevant for applications such as active learning (Hernández-Lobato & Adams, 2015) and reinforcement learning (Riquelme et al., 2018).

However, BNNs face two major issues: 1) the intractability of posterior inference and 2) the difficulty of choosing good Bayesian priors. While the former has been addressed in an extensive body of literature on variational inference (e.g. Blundell et al., 2015; Blei et al., 2016; Mishkin et al., 2018; Liu & Wang, 2016), the latter has only received limited attention (Vladimirova et al., 2019; Ghosh & Doshi-Velez, 2017). Choosing an informative prior for BNNs is particularly difficult due to the high-dimensional and hardly interpretable parameter space of NNs. Due to the lack of good alternatives, often a zero-centered, isotropic Gaussian is used, reflecting (almost) no a priori knowledge about the problem at hand. This does not only lead to poor generalization when data is scarce, but also renders the Bayesian uncertainty estimates poorly calibrated (Kuleshov et al., 2018).

Meta-learning (Schmidhuber, 1987; Thrun & Pratt, 1998) acquires inductive bias in a data-driven way, thus, constituting an alternative route for addressing this issue. In particular, meta-learners attempt to extract shared (prior) knowledge from a set of related learning tasks (i.e., datasets), aiming to learn in the face of a new, related task. Our work develops a principled and scalable algorithm for meta-learning BNN priors. We build on the PAC-Bayesian framework (McAllester, 1999), a methodology from statistical learning theory for deriving generalization bounds. Previous PAC-Bayesian bounds for meta-learners (Pentina & Lampert, 2014; Amit & Meir, 2018) require solving a difficult optimization problem, involving the optimization of the prior as well as multiple variational posteriors in a nested manner. Aiming to overcome this issue, we present a *PAC-Bayesian bound that does not rely on nested optimization* and, unlike (Rothfuss et al., 2020), can be *tractably optimized for BNNs*. This makes the resulting meta-learner, referred to as *PACOH-NN*, not only much more *computationally efficient and scalable* than previous approaches for meta-learning BNN priors (Amit & Meir, 2018), but also *agnostic to the choice of approximate posterior inference method* which allows us to combine it freely with recent advances in MCMC (e.g. Chen et al., 2014) or variational inference (e.g. Wang et al., 2019).

Our experiments demonstrate that the computational advantages of *PACOH-NN* do not result in degraded predictive performance. In fact, across several regression and classification environments, *PACOH-NN* achieves a *comparable or better predictive accuracy* than several popular meta-learning approaches, while *improving the quality of the uncertainty estimates*. Finally, we showcase how meta-learned *PACOH-NN* priors can be used in a real-world bandit task concerning the development of vaccines, suggesting that many other challenging real-world problems may benefit from our approach.

## 2 RELATED WORK

**Bayesian Neural Networks.** The majority of research on BNNs focuses on approximating the intractable posterior distribution (Graves, 2011; Blundell et al., 2015; Liu & Wang, 2016; Wang et al., 2019). In particular, we employ the approximate inference method of Liu & Wang (2016). Another crucial question is how to select a good BNN prior (Vladimirova et al., 2019). While the majority of work (e.g. Louizos & Welling, 2016; Huang et al., 2020) employs a simple zero-centered, isotropic Gaussian, Ghosh & Doshi-Velez (2017) and Pearce et al. (2020) have proposed other prior distributions for BNNs. In contrast, we go the alternative route of choosing priors in a data-driven way.

**Meta-learning.** A range of popular methods in meta-learning attempt to learn the "learning program" in form of a recurrent model (Hochreiter et al., 2001; Andrychowicz et al., 2016; Chen et al., 2017), learn an embedding space shared across tasks (Snell et al., 2017; Vinyals et al., 2016) or the initialization of a NN such that it can be quickly adapted to new tasks (Finn et al., 2017; Nichol et al., 2018; Rothfuss et al., 2019b). A group of recent methods also uses probabilistic modeling to also allow for uncertainty quantification (Kim et al., 2018; Finn et al., 2018; Garnelo et al., 2018). Although the mentioned approaches are able to learn complex inference patterns, they rely on settings where meta-training tasks are abundant and fall short of performance guarantees. In contrast, we provide a formal assessment of the generalization properties of our algorithm. Moreover, *PACOH-NN* allows for principled uncertainty quantification, including separate treatment of epistemic and aleatoric uncertainty. This makes it particularly useful for sequential decision algorithms (Lattimore & Szepesvari, 2020).

**PAC-Bayesian theory.** Previous work presents generalization bounds for randomized predictors, assuming a prior to be given exogenously (McAllester, 1999; Catoni, 2007; Germain et al., 2016; Alquier et al., 2016). More recent work explores data-dependent priors (Parrado-Hernandez et al., 2012; Dziugaite & Roy, 2016) or extends previous bounds to the scenario where priors are meta-learned (Pentina & Lampert, 2014; Amit & Meir, 2018). However, these meta-generalization bounds are hard to minimize as they leave both the hyper-posterior and posterior unspecified, which leads to nested optimization problems. Our work builds on the results of Rothfuss et al. (2020) who introduce the methodology to derive the closed-form solution of the PAC-Bayesian meta-learning problem. However, unlike ours, their approach suffers from (asymptotically) non-vanishing terms in the bounds and relies on a closed-form solution of the marginal log-likelihood. By contributing a numerically stable score estimator for the generalized marginal log-likelihood, we are able to overcome such limitations, making PAC-Bayesian meta-learning both tractable and scalable for a much larger array of models, including BNNs.

## 3 BACKGROUND: THE PAC-BAYESIAN FRAMEWORK

**Bayesian Neural Networks.** Consider a supervised learning task with data $S = \{(x_j, y_j)\}_{j=1}^m$ drawn from unknown distribution $\mathcal{D}$. In that, $\mathbf{X} = \{x_j\}_{j=1}^m \in \mathcal{X}^m$ denotes training inputs and $\mathbf{Y} = \{y_j\}_{j=1}^m \in \mathcal{Y}^m$ the targets. For brevity, we also write $z_j := (x_j, y_j) \in \mathcal{Z}$. Let $h_\theta : \mathcal{X} \to \mathcal{Y}$ be a function parametrized by a NN with weights $\theta \in \Theta$. Using the NN mapping, we define a conditional distribution $p(y|x, \theta)$. For regression, we set $p(y|x, \theta) = \mathcal{N}(y|h_\theta(x), \sigma^2)$, where $\sigma^2$ is the observation noise variance. For classification, we choose $p(y|x, \theta) = \text{Categorical}(\text{softmax}(h_\theta(x)))$. For Bayesian inference, one presumes a prior distribution $p(\theta)$ over the model parameters $\theta$ which is combined with the training data $S$ into a posterior distribution $p(\theta|\mathbf{X}, \mathbf{Y}) \propto p(\theta)p(\mathbf{Y}|\mathbf{X}, \theta)$. For unseen test data points $x^*$, we form the predictive distribution as $p(y^*|x^*, \mathbf{X}, \mathbf{Y}) = \int p(y^*|x^*, \theta)p(\theta|\mathbf{X}, \mathbf{Y})d\theta$.

The Bayesian framework presumes partial knowledge of the data-generating process in form of a prior distribution. However, due to the practical difficulties in choosing an appropriate BNN prior, the prior is typically strongly misspecified (Syring & Martin, 2018). As a result, modulating the influence of the prior relative to the likelihood during inference typically improves the empirical performance of BNNs and is thus a common practice (Wenzel et al., 2020). Using such a "tempered" posterior $p_\tau(\theta|\mathbf{X}, \mathbf{Y}) \propto p(\theta)p(\mathbf{Y}|\mathbf{X}, \theta)^\tau$ with $\tau > 0$ is also referred to as *generalized Bayesian learning* (Guedj, 2019).

**The PAC-Bayesian Framework.** In the following, we introduce the most relevant concepts of PAC-Bayesian learning theory. For more details, we refer to Guedj (2019). Given a loss function $l : \theta \times \mathcal{Z} \to \mathbb{R}$, we typically want to minimize the *generalization error* $\mathcal{L}(\theta, \mathcal{D}) = \mathbb{E}_{z^* \sim \mathcal{D}} \, l(\theta, z^*)$. Since $\mathcal{D}$ is unknown, the *empirical error* $\hat{\mathcal{L}}(\theta, S) = \frac{1}{m} \sum_{i=1}^{m} l(\theta, z_i)$ is usually employed, instead. In the PAC-Bayesian framework, we are concerned with *randomized predictors*, i.e., probability measures on the parameter space $\Theta$, allowing us to reason about *epistemic uncertainty*. In particular, we consider two such probability measures, a *prior* $P \in \mathcal{M}(\Theta)$ and a *posterior* $Q \in \mathcal{M}(\Theta)$. In here, by $\mathcal{M}(\Theta)$, we denote the set of all probability measures on $\Theta$. While in Bayesian inference, the prior and posterior are tightly coupled through Bayes' theorem, the PAC-Bayesian framework only requires the prior to be independent of the data $S$. Using the definitions above, the so-called *Gibbs error* for a randomized predictor $Q$ is defined as $\mathcal{L}(Q, \mathcal{D}) = \mathbb{E}_{h \sim Q} \, \mathcal{L}(h, \mathcal{D})$. Similarly, we define its empirical counterpart as $\hat{\mathcal{L}}(Q, S) = \mathbb{E}_{h \sim Q} \, \hat{\mathcal{L}}(h, S)$. The PAC-Bayesian framework provides upper bounds for the unknown *Gibbs error* in the following form:

**Theorem 1.** *(Alquier et al., 2016) Given a data distribution $\mathcal{D}$, a prior $P \in \mathcal{M}(\Theta)$, a confidence level $\delta \in (0, 1]$, with probability at least $1 - \delta$ over samples $S \sim \mathcal{D}^m$, we have:*

$$\forall Q \in \mathcal{M}(\Theta) : \quad \mathcal{L}(Q, \mathcal{D}) \le \hat{\mathcal{L}}(Q, S) + \frac{1}{\sqrt{m}} \left[ D_{KL}(Q||P) + \ln \frac{1}{\delta} + \Psi(\sqrt{m}) \right] \quad (1)$$

*where $\Psi(\sqrt{m}) = \ln \mathbb{E}_{\theta \sim P} \mathbb{E}_{S \in \mathcal{D}^m} \exp \left[ \sqrt{m} \left( \mathcal{L}(\theta, \mathcal{D}) - \hat{\mathcal{L}}(\theta, S) \right) \right].$*

In that, $\Psi(\sqrt{m})$ is a log moment generating function that quantifies how strong the empirical error deviates from the Gibbs error. By making additional assumptions about the loss function $l$, we can bound $\Psi(\sqrt{m})$ and thereby obtain tractable bounds. For instance, if $l(\theta, z)$ is *bounded* in $[a, b]$, we obtain $\Psi(\sqrt{m}) \le ((b-a)^2)/8$ by Hoeffding's lemma. For unbounded loss functions, it is common to assume bounded moments. For instance, a loss is considered *sub-gamma* with variance factor $s^2$ and scale parameter $c$, under a prior $P$ and data distribution $\mathcal{D}$, if its deviations from the mean can be characterized by random variable $V := \mathcal{L}(h, \mathcal{D}) - l(h, z)$ whose moment generating function is upper bounded by that of a Gamma distribution $\Gamma(s, c)$ (Boucheron et al., 2013). In such case, we obtain $\Psi(\sqrt{m}) \le s^2/(2 - \frac{2c}{\sqrt{m}})$.

**Connecting the PAC-Bayesian framework and generalized Bayesian learning.** In PAC-Bayesian learning we aim to find the posterior that minimizes the bound in (1) which is in general a challenging optimization problem over the space of measures $\mathcal{M}(\Theta)$. However, to our benefit, it can be shown that the *Gibbs posterior* is the probability measure that minimizes (1). For details we refer to Lemma 2 in the Appendix or Catoni (2007) and (Germain et al., 2016). In particular, this gives us

$$Q^*(\theta) := \underset{Q \in \mathcal{M}(\Theta)}{\arg\min} \sqrt{m} \hat{\mathcal{L}}(Q, S) + D_{KL}(Q||P) = P(\theta) e^{-\sqrt{m} \hat{\mathcal{L}}(\theta, S)} / Z(S, P), \quad (2)$$

where $Z(S, P)$ is a normalization constant. In a probabilistic setting, our loss function is the negative log-likelihood, i.e. $l(\theta, z_i) := -\log p(z_i|\theta)$. In this case, the optimal Gibbs posterior coincides with the *generalized Bayesian posterior* $Q^*(\theta; P, S) \propto P(\theta) p(S|\theta)^{1/\sqrt{m}} / Z(S, P)$ where $Z(S, P) = \int_\Theta P(\theta) p(S|\theta)^{1/\sqrt{m}} \, d\theta$ is the *generalized marginal likelihood* of the data sample $S$.

## 4 PAC-BAYESIAN BOUNDS FOR META-LEARNING

This section describes the PAC-Bayesian meta-learning setup and discusses how we can obtain generalization bounds that can be transformed into practically useful meta-learning objectives. In that, we draw on the methodology of Rothfuss et al. (2020) which allows us to derive a closed form solution of the PAC-Bayesian meta-learning problem.

In the standard supervised learning setup (see Sec. 3), the learner has prior knowledge in the form of a prior distribution $P$, given exogenously, over the hypothesis space $\mathcal{H}$. When the learner faces a new task, it uses the evidence, provided as a dataset $S$, to update the prior into a posterior distribution $Q$. We formalize such a *base learner* $Q(S, P)$ as a mapping $Q : \mathcal{Z}^m \times \mathcal{M}(\mathcal{H}) \to \mathcal{M}(\mathcal{H})$ that takes in a dataset and prior and outputs a posterior.

In contrast, in meta-learning we aim to acquire such a prior $P$ in a *data-driven manner*, that is, by consulting a set of $n$ statistically related learning tasks $\{\tau_1, ..., \tau_n\}$. We follow the setup of Baxter (2000) in which all tasks $\tau_i := (\mathcal{D}_i, S_i)$ share the same data domain $\mathcal{Z} := \mathcal{X} \times \mathcal{Y}$, parameter space $\Theta$ and loss function $l(\theta, z)$, but may differ in their (unknown) data distributions $\mathcal{D}_i$ and the number of points

$m_i$ in the corresponding dataset $S_i \sim \mathcal{D}_i^{m_i}$. For our theoretical expositions, we assume w.l.o.g that $m = m_i \; \forall i$. Further, each task $\tau_i \sim \mathcal{T}$ is drawn i.i.d. from an *environment* $\mathcal{T}$, a probability distribution over data distributions and datasets. The goal is to extract knowledge from the observed datasets which can then be used as a prior for learning on new target tasks $\tau \sim \mathcal{T}$ (Amit & Meir, 2018). To extend the PAC-Bayesian analysis to the meta-learning setting, we again consider the notion of probability distributions over hypotheses/parameters. However, while the object of learning has previously been the NN parameters $\theta$, it is now the prior distribution $P \in \mathcal{M}(\Theta)$. Accordingly, the meta-learner presumes a *hyper-prior* $\mathcal{P} \in \mathcal{M}(\mathcal{M}(\mathcal{H}))$, i.e., a distribution over priors $P$. Combining the hyper-prior $\mathcal{P}$ with the datasets $S_1, ..., S_n$ from multiple tasks, the meta-learner then outputs a *hyper-posterior* $\mathcal{Q}$ over priors. The hyper-posterior's performance/quality is measured in form of the expected Gibbs error when sampling priors $P$ from $\mathcal{Q}$ and applying the base learner, the so-called *transfer-error:*

$$\mathcal{L}(\mathcal{Q}, \mathcal{T}) := \mathbb{E}_{P \sim \mathcal{Q}} \left[ \mathbb{E}_{(\mathcal{D},m) \sim \mathcal{T}} \left[ \mathbb{E}_{S \sim \mathcal{D}^m} \left[ \mathcal{L}(Q(S,P), \mathcal{D}) \right] \right] \right]$$

While the transfer error is unknown in practice, we can estimate it using the *empirical multi-task error*

$$\hat{\mathcal{L}}(\mathcal{Q}, S_1, ..., S_n) := \mathbb{E}_{P \sim \mathcal{Q}} \left[ \frac{1}{n} \sum_{i=1}^{n} \hat{\mathcal{L}}(Q(S_i, P), S_i) \right] .$$

Similar to the PAC-Bayesian guarantees for single task learning, we can bound the transfer error by its empirical counterpart $\hat{\mathcal{L}}(\mathcal{Q}, S_1, ..., S_n)$ plus several tractable complexity terms:

**Theorem 2.** *Let $Q : \mathcal{Z}^m \times \mathcal{M}(\mathcal{H}) \to \mathcal{M}(\mathcal{H})$ be a base learner and $\mathcal{P} \in \mathcal{M}(\mathcal{M}(\mathcal{H}))$ some fixed hyper-prior. For all hyper-posteriors $\mathcal{Q} \in \mathcal{M}(\mathcal{M}(\mathcal{H}))$ and $\delta \in (0, 1]$,*

$$\mathcal{L}(\mathcal{Q}, \mathcal{T}) \le \hat{\mathcal{L}}(\mathcal{Q}, S_1, ..., S_n) + \left( \frac{1}{n\sqrt{m}} + \frac{1}{\sqrt{n}} \right) D_{KL}(\mathcal{Q} || \mathcal{P})$$

$$+ \frac{1}{n} \sum_{i=1}^{n} \frac{1}{\sqrt{m}} \mathbb{E}_{P \sim \mathcal{Q}} \left[ D_{KL}(Q(S_i, P) || P) \right] + C(\delta, n, m) \tag{3}$$

*holds with probability $1 - \delta$. If the loss function is bounded, that is $l : \mathcal{H} \times \mathcal{Z} \to [a, b]$, the above inequality holds with $C(\delta, n, m) = \frac{(b_k - a_k)^2}{8\sqrt{m}} + \frac{(b_k - a_k)^2}{8\sqrt{n}} - \frac{1}{\sqrt{n}} \ln \delta$. If the loss function is sub-gamma with variance factor $s_I^2$ and scale parameter $c_I$ for data distributions $\mathcal{D}_i$ and $s_{II}^2$, $c_{II}$ for the task distribution $\mathcal{T}$, the inequality holds with $C(\delta, n, m) = \frac{s_I^2}{2(\sqrt{m} - c_I)} + \frac{s_{II}^2}{2(\sqrt{n} - c_{II})} - \frac{1}{\sqrt{n}} \ln \delta$ .*

Under bounded loss assumption, Theorem 2 provides a structurally similar, but tighter bound than Pentina & Lampert (2014) and Rothfuss et al. (2020). In particular, by using a improved proof methodology, we are able to forgo a union bound argument, allowing us to reduce the negative influence of confidence parameter $\delta$. Compared to Rothfuss et al. (2020), the $D_{KL}(\mathcal{Q}, \mathcal{P})$ term has an improved decay rate, that is, $1/(n\sqrt{m}) + 1/\sqrt{n}$ as opposed to $1/\sqrt{m} + 1/\sqrt{n}$. Importantly, the bound in (3) is consistent, i.e. $C(\delta, n, m) \to 0$ as $n, m \to \infty$. Unlike Pentina & Lampert (2014) and Amit & Meir (2018), the theorem also provides guarantees for *unbounded* loss functions under moment constraints (see Appendix A.1 for details). This makes Theorem 2 particularly relevant for probabilistic models such as BNNs in which the loss function coincides with the inherently unbounded negative log-likelihood.

Amit & Meir (2018) propose to meta-learn NN priors by directly minimizing a bound similar to (3). However, the posterior inference for BNNs, i.e. obtaining $Q_i = Q(S_i, P)$, is a stochastic optimization problem in itself whose solution in turn depends on $P$. Hence, minimizing such meta-level bound w.r.t. $P$ constitutes a computationally infeasible two-level optimization problem. To circumvent this issue, they jointly optimize $P$ and $n$ approximate posteriors $Q_i$ that correspond to the different datasets $S_i$, leading to an unstable and poorly scalable meta-learning algorithm.

To overcome these issues, we employ the methodology of Rothfuss et al. (2020) and assume the Gibbs posterior $Q^*(S_i, P)$ as a base learner. As discussed in Section 3, $Q^*(S_i, P)$ does not only constitute a generalized Bayesian posterior but also minimizes the PAC-Bayesian bound. Thus, the resulting bound in Corollary 1 is tighter than (3). More importantly, the bound can be stated in terms of the partition function $Z(S_i, P)$ which allows us to forgo the explicit reliance on the task posteriors $Q_i$. This makes the bound much easier to optimize as a meta-learning objective than previous bounds (e.g. Pentina & Lampert, 2014; Amit & Meir, 2018), since it no longer constitutes a two-level optimization problem. Moreover, it renders the corresponding meta-learner agnostic

to the choice of approximate inference method used to approach the Gibbs/Bayes posterior $Q^*(S, P)$ when performing inference on a new task.

**Corollary 1.** *When choosing the Gibbs posterior $Q^*(S_i, P) := P(\theta) \exp(-\sqrt{m}\hat{\mathcal{L}}(S_i, \theta))/Z(S_i, P)$ as a base learner, under the same assumptions as in Theorem 2, it holds with probability $1 - \delta$ that*

$$\mathcal{L}(\mathcal{Q}, \mathcal{T}) \leq -\frac{1}{n} \sum_{i=1}^{n} \frac{1}{\sqrt{m}} \mathbb{E}_{P \sim \mathcal{Q}} [\ln Z(S_i, P)] + \left( \frac{1}{\sqrt{n}} + \frac{1}{n\sqrt{m}} \right) D_{KL}(\mathcal{Q}||\mathcal{P}) + C(\delta, n, m) \quad (4)$$

*wherein $Z(S_i, P) = \mathbb{E}_{\theta \sim P} \left[ \exp(-\sqrt{m}\hat{\mathcal{L}}(S_i, \theta)) \right]$ is the generalized marginal log-likelihood.*

A natural way to obtain a PAC-Bayesian meta-learning algorithm could be to minimize (4) w.r.t. $\mathcal{Q}$. Though, in general, this is a hard problem since it would require a minimization over $\mathcal{M}(\mathcal{M}(\mathcal{H}))$, the space of all probability measures over priors. Following Rothfuss et al. (2020), we exploit once more the insight that the minimizer of (4) can be written as Gibbs distribution (c.f. Lemma 2), allowing us to to derive such minimizing hyper-posterior $\mathcal{Q}^*$, i.e. the *PACOH*, in closed form:

**Proposition 1.** *(PAC-Optimal Hyper-Posterior) Given a hyper-prior $\mathcal{P}$ and datasets $S_1, ..., S_n$, the hyper-posterior $\mathcal{Q}$ that minimizes the PAC-Bayesian meta-learning bound in (4) is given by*

$$\mathcal{Q}^*(P) = \frac{\mathcal{P}(P) \exp \left( \frac{1}{\sqrt{nm}+1} \sum_{i=1}^{n} \ln Z(S_i, P) \right)}{Z^{II}(S_1, ..., S_n, \mathcal{P})} \quad (5)$$

*with the partition function defined as $Z^{II} = \mathbb{E}_{P \sim \mathcal{P}} \left[ \exp \left( \frac{1}{\sqrt{nm}+1} \sum_{i=1}^{n} \ln Z(S_i, P) \right) \right]$ .*

We refer to $\mathcal{Q}^*(P)$ as *PAC-optimal* since, among all meta-learners, it gives us the best possible PAC-Bayesian guarantees induced by Theorem 2.

## 5 PACOH-NN: A SCALABLE ALGORITHM FOR LEARNING BNN PRIORS

We discuss our main contribution, that is, how to translate the *PACOH* (Prop. 1) into a practical algorithm for meta-learning BNN priors. To this end, we first specify various components of the generic meta-learning setup presented in Sec. 4 and then discuss how to obtain a tractable approximation of $\mathcal{Q}^*$.

**The setup.** First, we define a family of priors $\{P_\phi : \phi \in \Phi\}$ over the NN parameters $\theta$. For computational convenience, we employ Gaussian priors with diagonal covariance matrix, i.e. $P_\phi = \mathcal{N}(\mu_P, \text{diag}(\sigma_P^2))$ with $\phi := (\mu_P, \log \sigma_P)$. Note that we represent $\sigma_P$ in the log-space to avoid additional positivity constraints. In fact, any parametric distribution such as normalizing flows (Rezende & Mohamed, 2015) that allows for re-parametrized sampling and has a tractable log-density could be used. As typical in the Bayesian framework, our loss function is the negative log-likelihood, i.e. $l(\theta, z) = -\ln p(y|x, \theta)$ for which we assume an additive Gaussian noise model $p(y|x, \theta) = \mathcal{N}(y; h_\theta(x), \sigma_y^2)$ in regression and a categorical softmax distribution in case of classification. Moreover, we use a zero-centered, spherical Gaussian $\mathcal{P} := \mathcal{N}(0, \sigma_\mathcal{P}^2 I)$ as a hyper-prior over the parameters $\phi$ that specify the prior. In our setup, the hyper-prior acts a form of meta-level regularization that penalizes complex priors.

**Approximating the hyper-posterior.** Given the hyper-prior and (level-I) log-partition function $\ln Z(S_i, P)$, we can compute the PACOH $\mathcal{Q}^*$ up to the normalization constant $Z^{II}$. Such a setup lends itself to approximate inference methods (Blei et al., 2016). In particular, we employ Stein Variational Gradient Descent (SVGD) (Liu & Wang, 2016) which approximates $\mathcal{Q}^*$ as a set of particles $\hat{\mathcal{Q}} = \{\phi_1, ..., \phi_K\}$. Initially, the particles $\phi_K \sim \mathcal{P}$ (i.e. the priors' parameters) are sampled randomly. Subsequently, the method iteratively transports the set of particles to match $\mathcal{Q}^*$, by applying a form of functional gradient descent that minimizes $D_{KL}(\hat{\mathcal{Q}}|\mathcal{Q}^*)$ in the reproducing kernel Hilbert space induced by a kernel function $k(\cdot, \cdot)$. In each iteration, the particles are updated by $\phi_k \leftarrow \phi_k + \eta\psi^*(\phi_k)$ with the step size $\eta$ and

$$\psi^*(\phi) = \frac{1}{K} \sum_{k'=1}^{K} \left[ k(\phi_{k'}, \phi) \nabla_{\phi_{k'}} \ln \mathcal{Q}^*(\phi_{k'}) + \nabla_{\phi_{k'}} k(\phi_{k'}, \phi) \right]. \quad (6)$$

In that, $\nabla_{\phi_k} \ln \mathcal{Q}^*(\phi_k) = \nabla_{\phi_k} \ln \mathcal{P}(\phi_k) + \frac{1}{\sqrt{nm}+1} \sum_{i=1}^{n} \nabla_{\phi_k} \ln Z(S_i, P_{\phi_k})$ is the score of $\mathcal{Q}^*$.

**Approximating the generalized marginal log-likelihood.** The last remaining issue towards a viable meta-learning algorithm is the intractable generalized marginal likelihood $\ln Z(S_i, P_\phi) =$

(a) BNN with isotropic, zero-centered Gaussian prior     (b) BNN with meta-learned *PACOH-NN* prior

Figure 1: BNN posterior predictions with (a) standard Gaussian prior vs. (b) meta-learned prior. Meta-learning with *PACOH-NN-SVGD* was conducted on the *Sinusoids* environment.

$\ln \mathbb{E}_{\theta \sim P_\phi} e^{-\sqrt{m_i}\hat{\mathcal{L}}(\theta, S_i)}$. Estimating and optimizing $\ln Z(S_i, P_\phi)$ is not only challenging due to the high-dimensional expectation over $\Theta$ but also due to numerical instabilities inherent in computing $e^{-\sqrt{m_i}\hat{\mathcal{L}}(\theta, S_i)}$ when $m_i$ is large. Aiming to overcome these issues, we compute numerically stable Monte Carlo estimates of $\nabla_\phi \ln Z(S_i, P_{\phi_k})$ by combining the LogSumExp (LSE) with the re-parametrization trick (Kingma & Welling, 2014). In particular, we draw $L$ samples $\theta_l := f(\phi, \epsilon_l) = \mu_P + \sigma_P \odot \epsilon_l$, $\epsilon_l \sim N(0, I)$ and compute the forward pass as follows:

$$\ln \tilde{Z}(S_i, P_\phi) := \ln \frac{1}{L} \sum_{l=1}^{L} e^{-\sqrt{m_i}\hat{\mathcal{L}}(\theta_l, S_i)} = \text{LSE}_{l=1}^{L}\left(-\sqrt{m_i}\hat{\mathcal{L}}(\theta_l, S_i)\right) - \ln L , \ \theta_l \sim P_\phi \quad (7)$$

The corresponding gradients follow as softmax-weighted average of score gradients:

$$\nabla_\phi \ln \tilde{Z}(S_i, P_\phi) = -\sqrt{m_i} \sum_{l=1}^{L} \underbrace{\frac{e^{-\sqrt{m_i}\hat{\mathcal{L}}(\theta_l, S_i)}}{\sum_{l=1}^{L} e^{-\sqrt{m_i}\hat{\mathcal{L}}(\theta_l, S_i)}}}_{\text{softmax}} \underbrace{\nabla_\phi f(\phi, \epsilon_l)^\top}_{\substack{\text{re-param.} \\ \text{Jacobian}}} \underbrace{\nabla_{\theta_l}\hat{\mathcal{L}}(\theta_l, S_i)}_{\text{score}} \quad (8)$$

Note that $\ln \tilde{Z}(S_i, P_\phi)$ is a consistent but not an unbiased estimator of $\ln Z(S_i, P_\phi)$. The following proposition ensures us that we still minimize a valid bound (see Appx. B.3 for details).

**Proposition 2.** *In expectation, replacing $\ln Z(S_i, P_\phi)$ in (4) by the estimate $\ln \tilde{Z}(S_i, P)$ still yields a valid upper bound of the transfer error $\mathcal{L}(\mathcal{Q}, \mathcal{T})$ for any $L \in \mathbb{N}$.*

Moreover, by the law of large numbers, we have that $\ln \tilde{Z}(S_i, P) \xrightarrow{\text{a.s.}} \ln Z(S_i, P)$ as $L \to \infty$, that is, for large sample sizes $L$, we recover the original PAC-Bayesian bound in (4). In the opposite edge case, i.e. $L = 1$, the boundaries between tasks vanish meaning that the meta-training data $\{S_1, ..., S_n\}$ is treated as if it were one large dataset $\bigcup_i S_i$ (see Appx. B.3 for further discussion).

**The algorithm.** Algorithm 1 in Appendix B summarizes the proposed meta-learning method which we henceforth refer to as *PACOH-NN*. To estimate the score $\nabla_{\phi_{k'}} \ln \mathcal{Q}^*(\phi_{k'})$ in (6), we can even use mini-batching on the task level. This mini-batched version, outlined in Algorithm 2, maintains $K$ particles to approximate the hyper-posterior, and in each forward step samples $L$ NN-parameters (of dimensionaly $|\Theta|$) per prior particle that are deployed on a mini-batch of $n_{bs}$ tasks to estimate the score of $\mathcal{Q}^*$. As a result, the total space complexity is in the order of $\mathcal{O}(|\Theta|K + L)$ and the computational complexity of the algorithm for a single update (c.f. (6)) is $\mathcal{O}(K^2 + KLn_{bs})$.

A key advantage of *PACOH-NN* over previous methods for meta-learning BNN priors (e.g. Pentina & Lampert, 2014; Amit & Meir, 2018) is that it turns the previously nested optimization problem into a much simpler stochastic optimization problem. This makes meta-learning not only much more stable but also more scalable. In particular, we do not need to explicitly compute / maintain the task posteriors $Q_i$ and can do mini-batching over tasks. As a result, the space and compute complexity do not depend on the number of tasks $n$. In contrast, *MLAP* (Amit & Meir, 2018) has a memory footprint of $\mathcal{O}(|\Theta|n)$ making meta-learning prohibitive for more than 50 tasks.

A central feature of *PACOH-NN* is that is comes with principled meta-level regularization in form of the hyper-prior $\mathcal{P}$ which combats overfitting to the meta-training tasks (Qin et al., 2018). As we show in our experiments, this allows us to successfully perform meta-learning with as little as 5 tasks. This is unlike the the majority of the popular meta-learners (Finn et al., 2017; Kim et al., 2018; Garnelo et al., 2018, e.g.) which rely on a large number of tasks to generalize well on the meta-level (Qin et al., 2018; Rothfuss et al., 2020).

## 6 EXPERIMENTS

We empirically evaluate the method introduced in Section 5, in particular, two variants of the algorithm: *PACOH-NN-SVGD* with $K = 5$ priors as particles and the edge case $K = 1$ which

| | Cauchy | SwissFel | Physionet-GCS | Physionet-HCT | Berkeley-Sensor |
|---|---|---|---|---|---|
| BNN (Liu & Wang, 2016) | $0.327 \pm 0.008$ | $0.529 \pm 0.022$ | $2.664 \pm 0.274$ | $3.938 \pm 0.869$ | $0.109 \pm 0.004$ |
| PACOH-NN-SVGD (ours) | $\mathbf{0.195 \pm 0.001}$ | $\mathbf{0.372 \pm 0.002}$ | $1.561 \pm 0.061$ | $\mathbf{2.405 \pm 0.017}$ | $\mathbf{0.043 \pm 0.001}$ |
| PACOH-NN-MAP (ours) | $0.202 \pm 0.003$ | $0.375 \pm 0.004$ | $1.564 \pm 0.200$ | $2.480 \pm 0.042$ | $0.047 \pm 0.001$ |
| PACOH-GP (Rothfuss et al., 2020) | $0.209 \pm 0.008$ | $0.376 \pm 0.024$ | $\mathbf{1.498 \pm 0.081}$ | $\mathbf{2.361 \pm 0.047}$ | $0.065 \pm 0.005$ |
| MLAP-M (Amit & Meir, 2018) | $0.219 \pm 0.002$ | $0.492 \pm 0.009$ | $2.232 \pm 0.261$ | $2.541 \pm 0.140$ | $0.052 \pm 0.003$ |
| MLAP-S (Amit & Meir, 2018) | $0.219 \pm 0.004$ | $0.486 \pm 0.026$ | $2.009 \pm 0.248$ | $2.470 \pm 0.039$ | $0.050 \pm 0.005$ |
| FOMAML (Nichol et al., 2018) | $0.260 \pm 0.007$ | $0.897 \pm 0.071$ | $2.545 \pm 0.615$ | $\mathbf{2.408 \pm 0.064}$ | $0.059 \pm 0.005$ |
| MAML (Finn et al., 2017) | $0.219 \pm 0.004$ | $0.730 \pm 0.057$ | $1.895 \pm 0.141$ | $2.413 \pm 0.113$ | $\mathbf{0.045 \pm 0.003}$ |
| BMAML (Kim et al., 2018) | $0.225 \pm 0.004$ | $0.577 \pm 0.044$ | $1.894 \pm 0.062$ | $2.500 \pm 0.002$ | $0.073 \pm 0.014$ |
| NPs (Garnelo et al., 2018) | $0.224 \pm 0.008$ | $0.471 \pm 0.053$ | $2.056 \pm 0.209$ | $2.594 \pm 0.107$ | $0.079 \pm 0.007$ |

Table 1: Comparison of standard and meta-learning algorithms in terms of test RMSE in 5 meta-learning environments for regression. Reported are mean and standard deviation across 5 seeds.

| | Cauchy | SwissFel | Physionet-GCS | Physionet-HCT | Berkeley-Sensor |
|---|---|---|---|---|---|
| BNN (Liu & Wang, 2016) | $0.055 \pm 0.006$ | $0.085 \pm 0.008$ | $0.277 \pm 0.013$ | $0.307 \pm 0.009$ | $0.179 \pm 0.002$ |
| PACOH-NN-SVGD (ours) | $\mathbf{0.046 \pm 0.001}$ | $\mathbf{0.027 \pm 0.003}$ | $0.267 \pm 0.005$ | $0.302 \pm 0.003$ | $\mathbf{0.067 \pm 0.005}$ |
| PACOH-NN-MAP (ours) | $0.051 \pm 0.002$ | $0.031 \pm 0.003$ | $0.268 \pm 0.015$ | $0.306 \pm 0.003$ | $\mathbf{0.063 \pm 0.016}$ |
| PACOH-GP (Rothfuss et al., 2020) | $0.056 \pm 0.004$ | $0.038 \pm 0.006$ | $\mathbf{0.262 \pm 0.004}$ | $\mathbf{0.296 \pm 0.003}$ | $0.098 \pm 0.005$ |
| MLAP-M (Amit & Meir, 2018) | $0.088 \pm 0.004$ | $0.104 \pm 0.015$ | $0.339 \pm 0.012$ | $0.297 \pm 0.007$ | $0.077 \pm 0.010$ |
| MLAP-S (Amit & Meir, 2018) | $0.086 \pm 0.015$ | $0.090 \pm 0.021$ | $0.343 \pm 0.017$ | $0.344 \pm 0.016$ | $0.108 \pm 0.024$ |
| BMAML (Kim et al., 2018) | $0.061 \pm 0.007$ | $0.115 \pm 0.036$ | $0.279 \pm 0.010$ | $0.423 \pm 0.106$ | $0.161 \pm 0.013$ |
| NPs (Garnelo et al., 2018) | $0.057 \pm 0.009$ | $0.131 \pm 0.056$ | $0.299 \pm 0.012$ | $0.319 \pm 0.004$ | $0.210 \pm 0.007$ |

Table 2: Comparison of standard and meta-learning algorithms in terms of test calibration error in 5 meta-learning environments for regression. Reported are mean and standard deviation across 5 seeds.

coincides with a maximum-a-posteriori (MAP) approximation of $\mathcal{Q}^*$, thus referred to as *PACOH-NN-MAP*. In order to evaluate the quality of the meta-learned prior, i.e. *meta-testing*, we need to do approximate the BNN posterior $Q^*(S, P)$ for which we use SVGD with 5 particles, too.

Comparing it to various NN-based meta-learning approaches on various regression and classification environments, we demonstrate that *PACOH-NN* (i) outperforms previous meta-learning algorithms in terms of predictive accuracy, (ii) improves the quality of uncertainty estimates and (iii) is much more scalable than previous PAC-Bayesian meta-learners. Finally, we showcase how meta-learned *PACOH-NN* priors can be harnessed in a real-world bandit task concerning peptide-based vaccine development.

## 6.1 META-LEARNING BNN PRIORS FOR REGRESSION AND CLASSIFICATION

Figure 1 illustrates *BNN* predictions on a sinusoidal regression task with a standard Gaussian prior as well as a *PACOH-NN* prior meta-learned on sinusoidal functions of varying amplitude, phase and slope (details can be found in Appendix C.1). In Figure 1a we can see that the standard Gaussian prior provides poor inductive bias, not only leading to bad mean predictions away from the testing points but also to poor 95% confidence intervals (blue shaded areas). In contrast, the meta-learned *PACOH-NN* prior encodes useful inductive bias towards sinusoidal function shapes, leading to good predictions and uncertainty estimates, even in face of minimal training data (i.e. 1 training point).

**Meta-learning benchmark.** In the following, we present a comprehensive benchmark study. First, we use a *BNN* with a zero-centered, spherical Gaussian prior and SVGD posterior inference (Liu & Wang, 2016) as a baseline. Second, we compare our proposed approach against various popular meta-learning algorithms, including model-agnostic meta-learning (*MAML*) (Finn et al., 2017), its first-order version (*FOMAML*) (Nichol et al., 2018), Bayesian MAML (*BMAML*) (Kim et al., 2018) and two variants of the PAC-Bayesian approach by Amit & Meir (2018) (*MLAP*). For experiments with regression tasks, we also include into our comparison neural processes (*NPs*) (Garnelo et al., 2018) and the GP based meta-learner of Rothfuss et al. (2020) (*PACOH-GP*). The latter, is similar to our method as it also approximates a form of the PAC-optimal Hyper-Posterior with SVGD. However, unlike *PACOH-NN* it uses Gaussian Processes (GPs) as base learners and relies on a closed-form marginal log-likelihood. Among all, *MLAP* is the most similar to our approach as it is neural network based and minimizes PAC-Bayesian bounds of the transfer error. Though, unlike *PACOH-NN*, it relies on nested optimization of the task posteriors $Q_i$ and the hyper-posterior $\mathcal{Q}$.

**Regression environments.** In our experiments, we consider one synthetic and four real-world meta-learning environments for *regression*. As a synthetic environment we follow Rothfuss et al. (2020), employing a 2-dimensional mixture of *Cauchy* distributions plus random GP functions. As real-world environments, we use datasets corresponding to different calibration sessions of the Swiss Free Electron Laser (*SwissFEL*) (Milne et al., 2017; Kirschner et al., 2019b), as well as data from the *PhysioNet* 2012 challenge, which consists of time series of electronic health measurements from

|  | Accuracy | | Calibration error | |
|---|---|---|---|---|
|  | Omniglot 2-shot | Omniglot 5-shot | Omniglot 2-shot | Omniglot 5-shot |
| BNN (Liu & Wang, 2016) | $0.6709 \pm 0.006$ | $0.795 \pm 0.006$ | $0.173 \pm 0.009$ | $0.135 \pm 0.009$ |
| PACOH-NN-SVGD (ours) | $0.733 \pm 0.009$ | $\mathbf{0.885 \pm 0.090}$ | $\mathbf{0.094 \pm 0.004}$ | $0.091 \pm 0.010$ |
| PACOH-NN-MAP (ours) | $\mathbf{0.735 \pm 0.010}$ | $0.866 \pm 0.005$ | $0.099 \pm 0.009$ | $\mathbf{0.075 \pm 0.006}$ |
| MLAP-M (Amit & Meir, 2018) | $0.635 \pm 0.015$ | $0.804 \pm 0.0168$ | $0.108 \pm 0.008$ | $0.119 \pm 0.0193$ |
| MLAP-S (Amit & Meir, 2018) | $0.615 \pm 0.037$ | $0.700 \pm 0.0135$ | $0.129 \pm 0.018$ | $0.108 \pm 0.010$ |
| FO-MAML (Nichol et al., 2018) | $0.429 \pm 0.047$ | $0.590 \pm 0.010$ | N/A | N/A |
| MAML (Finn et al., 2017) | $0.571 \pm 0.018$ | $0.693 \pm 0.013$ | N/A | N/A |
| BMAML (Kim et al., 2018) | $0.651 \pm 0.008$ | $0.764 \pm 0.025$ | $0.132 \pm 0.007$ | $0.191 \pm 0.018$ |

Table 3: Comparison of meta-learning algorithms in terms of test accuracy and calibration error on the *Omniglot* environment with 2-shot and 5-shot 5-way-classification tasks.

intensive care patients (Silva et al., 2012), in particular the Glasgow Coma Scale (*GCS*) and the hematocrit value (*HCT*). Here, the different tasks correspond to different patients. Moreover, we employ the Intel Berkeley Research lab temperature sensor dataset (*Berkeley-Sensor*) (Madden, 2004) where the tasks require auto-regressive prediction of temperatures measurements corresponding to sensors installed in different locations of the building. Further details can be found in Appendix C.1.

Table 1 reports the results of our benchmark study in terms of the root mean squared error (RMSE) on unseen test tasks. Among the approaches, *PACOH-NN* consistently performs best or is among the best two methods, demonstrating that the introduced meta-learning framework is not only sound, but also endows us with an algorithm that works well in practice. Only for low-dimensional and small-scale regression environments like *Physionet*, we find that the GP-based meta-learner *PACOH-GP* which is build on a similar theoretical foundation as our method works better than *PACOH-NN*.

Further, we hypothesize that by acquiring the prior in a principled data-driven manner (e.g., with *PACOH-NN*), we can improve the quality of the BNN's uncertainty estimates. To investigate the effect of meta-learned priors on the uncertainty estimates of the BNN, we compute the probabilistic predictors' calibration error, reported in Table 2. The *calibration error* measures the discrepancy between predicted confidence regions and actual frequencies of test data in the respective areas (Kuleshov et al., 2018). Note that, since *MAML* only produces point predictions, the concept of calibration does not apply to it. First, we observe that meta-learning priors with *PACOH-NN* consistently improves the standard BNN's uncertainty estimates. For meta-learning environments where the task similarity is high, i.e. *SwissFel* and *Berkeley-Sensor*, the improvement is substantial. Surprisingly, while improving upon the standard BNN in terms of the RMSE, we find that *NPs* consistently yields worse-calibrated predictive distributions than the BNN without meta-learning. This may be due to meta-level overfitting as NPs lack any form of meta-level regularization (cf. Qin et al., 2018; Rothfuss et al., 2020).

**Classification environments.** We conduct experiments with the multi-task *classification* environment *Omniglot* (Lake et al., 2015), consisting of handwritten letters across 50 alphabets. Unlike previous work (e.g. Finn et al., 2017) we do not perform data-augmentation and do not re-combine letters of different alphabets, preserving the data's original structure. In particular, one task corresponds to 5-way classification of letters within an alphabet. This leaves us with much fewer tasks (i.e. 30 train and 20 test tasks), making the environment more challenging and more interesting for uncertainty quantification. This also allows us to include *MLAP* in the experiment which hardly scales to more than 50 tasks.

In Table 3, we report both the accuracy and calibration error for 2-shot and 5-shot classification on test tasks. Again, *PACOH-NN* yields the most accurate classification results and the lowest calibration error. Note that *MAML* fails to improve upon the standard BNN, i.e. demonstrating negative transfer. This is consistent with previous work (Qin et al., 2018; Rothfuss et al., 2020) raising concerns about overfitting to the meta-training tasks and observing that *MAML* requires a large number of tasks to generalize well. In contrast, by its very construction on meta-generalization

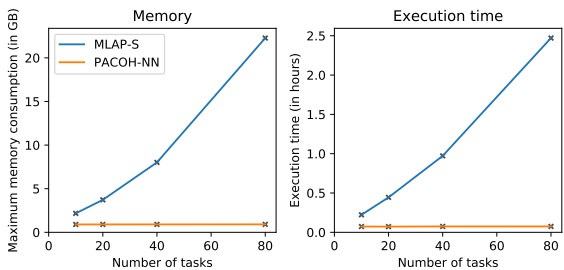

Figure 2: Scalability comparison of *PACOH-NN* and MLAP-S in memory footprint and compute time, as the number of meta-training task grows.

bounds, *PACOH-NN* is able to achieve positive transfer even when the meta-training tasks are diverse and small in number.

**Scalability.** Unlike the *MLAP* (Amit & Meir, 2018), *PACOH-NN* does not need to maintain posteriors $Q_i$ for the meta-training tasks and can use mini-batching on the task level. As a result, it is computationally much faster and more scalable than previous PAC-Bayesian meta-learners. This is reflected in its computation and memory complexity, discussed in Section 5. Figure 2 showcases this computational advantage during meta-training with *PACOH-NN-MAP* and *MLAP-S* in the *Sinusoids* environment with varying number of tasks, reporting the maximum memory requirements, as well as the training time. While *MLAP's* memory consumption and compute time grow linearly and becomes prohibitively large even for less than 100 tasks, *PACOH-NN* maintains a constant memory and compute load as the number of tasks grow.

## 6.2 META-LEARNING FOR BANDITS - VACCINE DEVELOPMENT

We showcase how a relevant real-world application such as vaccine design can benefit from our proposed method. In particular, the goal is to discover peptide sequences which bind to major histocompatibility complex class-I molecules (MHC-I). MHC-I molecules present fragments of proteins from within a cell to T-cells, allowing the immune system to distinguish between healthy and infected cells. Following the bandit setup of Krause & Ong (2011), each task corresponds to searching for maximally binding peptides, a vital step in the design of peptide-based vaccines. The tasks differ in their targeted MHC-I allele, i.e., correspond to different genetic variants of the MHC-I protein. We use data from Widmer et al. (2010) which contains the binding affinities ($IC_{50}$ values) of many peptide candidates to the MHC-I alleles. The peptide candidates are encoded as 45-dimensional feature vector and the binding affinities were standardized.

We use 5 alleles (tasks) to meta-learn a BNN prior with *PACOH-NN* and leave the most genetically dissimilar allele (A-6901) for our bandit task. In each iteration, the experimenter (i.e. bandit algorithm) chooses to test one peptide among the pool of more than 800 candidates and receives its binding affinity as a reward feedback. In particular, we employ UCB (Lattimore & Szepesvari, 2020) and Thompson-Sampling (TS) (Thompson, 1933) as bandit algorithms, comparing the BNN-based bandits with meta-learned prior (*PACOH-UCB/TS*) against a zero-centered Gaussian BNN prior (*BNN-UCB/TS*) and a Gaussian process (*GP-UCB*) (Srinivas et al., 2009).

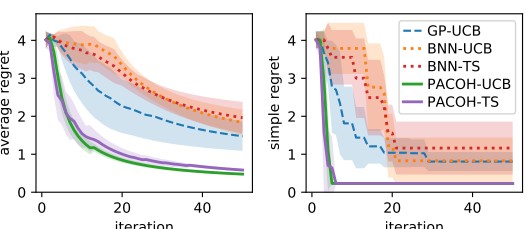

Figure 3: MHC-I peptide design task: Regret for different priors and bandit algorithms. A meta-learned *PACOH-NN* prior substantially improves the regret, compared to a standard BNN/GP prior.

Figure 3 reports the respective average regret and simple regret over 50 iterations. Unlike the bandit algorithms with standard BNN/GP prior, *PACOH-UCB/TS* reaches near optimal regret within less than 10 iterations and after 50 iterations still maintains a significant performance advantage. This highlights the importance of *transfer (learning)* for solving real-world problems and demonstrates the effectiveness of *PACOH-NN* to this end. While the majority of meta-learning methods rely on a large number of meta-training tasks (Qin et al., 2018), *PACOH-NN* allows us to achieve promising positive transfer, even in complex real-world scenarios with only a handful (in this case 5) tasks.

## 7 CONCLUSION

Based on PAC-Bayesian theory, we present a novel, scalable algorithm for meta-learning BNN priors, that overcomes previous issues of nested optimization, by employing the closed-form solution of the PAC-Bayesian meta-learning problem. Experiments show that our method, *PACOH-NN*, does not only come with computational advantages, but also achieves comparable or better predictive accuracy than several popular meta-learning approaches, while improving the quality of the uncertainty estimates – a key aspect of our approach. The benefits of our principled treatment of uncertainty – showcased in the real-world vaccine development bandit task – are particularly amenable to interactive machine learning systems. This makes the integration of *PACOH-NN* with Bayesian optimization and reinforcement learning a potentially promising avenue to pursue. While our experiments are limited to diagonal Gaussian priors and SVGD as approximate inference method, we hope that future work will build on the added flexibility of our framework to possibly explore more recent approaches in variational inference (e.g. Wang et al., 2019) or consider more expressive priors such as normalizing flows (Rezende & Mohamed, 2015).

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

APPENDIX

## A  PROOFS AND DERIVATIONS

### A.1  PROOF OF THEOREM 2

**Lemma 1.** *(Change of measure inequality) Let $f$ be a random variable taking values in a set $A$ and let $X_1, ..., X_l$ be independent random variables, with $X_k \in A$ with distribution $\mu_k$. For functions $g_k : A \times A \to \mathbb{R}, k = 1, ..., l$, let $\xi_k(f) = \mathbb{E}_{X_k \sim \mu_k}[g_k(f, X_k)]$ denote the expectation of $g_k$ under $\mu_k$ for any fixed $f \in A$. Then for any fixed distributions $\pi, \rho \in \mathcal{M}(A)$ and any $\lambda > 0$, we have that*

$$\mathbb{E}_{f \sim \rho}\left[\sum_{k=1}^{l} \xi_k(f) - g_k(f, X_k)\right] \leq \frac{1}{\lambda}\left(D_{KL}(\rho||\pi) + \ln \mathbb{E}_{f \sim \pi}\left[e^{\lambda\left(\sum_{k=1}^{l} \xi_k(f) - g_k(f, X_k)\right)}\right]\right). \tag{9}$$

**Proof of Theorem 2**  To prove the Theorem, we need to bound the difference between *transfer error* $\mathcal{L}(\mathcal{Q}, \mathcal{T})$ and the *empirical multi-task error* $\hat{\mathcal{L}}(\mathcal{Q}, S_1, ..., S_n)$. To this end, we introduce an intermediate quantity, the *expected multi-task error*:

$$\tilde{\mathcal{L}}(\mathcal{Q}, \mathcal{D}_1, ..., \mathcal{D}_n) = \mathbb{E}_{P \sim \mathcal{Q}}\left[\frac{1}{n}\sum_{i=1}^{n} \mathbb{E}_{S \sim D_i^{m_i}}\left[\mathcal{L}(Q(S, P), \mathcal{D}_i)\right]\right] \tag{10}$$

In the following we invoke Lemma 1 twice. First, in step 1, we bound the difference between $\tilde{\mathcal{L}}(\mathcal{Q}, \mathcal{D}_1, ..., \mathcal{D}_n)$ and $\hat{\mathcal{L}}(\mathcal{Q}, S_1, ..., S_n)$, then, in step 2, the difference between $\mathcal{L}(\mathcal{Q}, \mathcal{T})$ and $\tilde{\mathcal{L}}(\mathcal{Q}, \mathcal{D}_1, ..., \mathcal{D}_n)$. Finally, in step 3, we use a union bound argument to combine both results.

**Step 1 (Task specific generalization)**  First, we bound the generalization error of the observed tasks $\tau_i = (\mathcal{D}_i, m_i), i = 1, ..., n$, when using a learning algorithm $Q : \mathcal{M} \times \mathcal{Z}^{m_i} \to \mathcal{M}$, which outputs a posterior distribution $Q_i = Q(S_i, P)$ over hypotheses $\theta$, given a prior distribution $P$ and a dataset $S_i \sim \mathcal{D}_i^{m_i}$ of size $m_i$. In that, we define $\tilde{m} := (\sum_{i=1}^{n} m_i^{-1})^{-1}$ as the harmonic mean of sample sizes.

In particular, we apply Lemma 1 to the union of all training sets $S' = \bigcup_{i=1}^{n} S_i$ with $l = \sum_{i=1}^{n} m_i$. Hence, each $X_k$ corresponds to one data point, i.e. $X_k = z_{ij}$ and $\mu_k = \mathcal{D}_i$. Further, we set $f = (P, h_1, ..., h_n)$ to be a tuple of one prior and $n$ base hypotheses. This can be understood as a two-level hypothesis, wherein $P$ constitutes a hypothesis of the meta-learning problem and $h_i$ a hypothesis for solving the supervised task $\tau_i$. Correspondingly, we take $\pi = (\mathcal{Q}, Q^n) = \mathcal{P} \cdot \prod_{i=1}^{n} P$ and $\rho = (\mathcal{Q}, Q^n) = \mathcal{Q} \cdot \prod_{i=1}^{n} Q_i$ as joint two-level distributions and $g_k(f, X_k) = \frac{1}{nm_i} l(h_i, z_{ij})$ as summand of the empirical multi-task error. We can now invoke Lemma 1 to obtain that (11) and (14)

$$\frac{1}{n}\sum_{i=1}^{n} \mathbb{E}_{P \sim \mathcal{Q}}\left[\mathcal{L}(Q_i, \mathcal{D}_i)\right] \leq \frac{1}{n}\sum_{i=1}^{n} \mathbb{E}_{P \sim \mathcal{Q}}\left[\mathcal{L}(Q_i, S_i)\right] + \frac{1}{\lambda}\left(D_{KL}\left[(\mathcal{Q}, Q^n)||(\mathcal{P}, P^n)\right]\right.$$
$$\left. + \ln \mathbb{E}_{P \sim \mathcal{P}} \mathbb{E}_{h \sim P}\left[e^{\frac{\lambda}{n}\sum_{i=1}^{n}(\mathcal{L}(h, \mathcal{D}_i) - \hat{\mathcal{L}}(h, S_i))}\right]\right) \tag{11}$$

Using the above definitions, the KL-divergence term can be re-written in the following way:

$$D_{KL}\left[(\mathcal{Q}, Q^n)||(\mathcal{P}, P^n)\right] = \mathbb{E}_{P \sim \mathcal{Q}}\left[\mathbb{E}_{h \sim Q_i}\left[\ln \frac{\mathcal{Q}(P) \prod_{i=1}^{n} \prod Q_i(h)}{\mathcal{P}(P) \prod_{i=1}^{n} P(h)}\right]\right] \tag{12}$$

$$= \mathbb{E}_{P \sim \mathcal{Q}}\left[\ln \frac{\mathcal{Q}(P)}{\mathcal{P}(P)}\right] + \sum_{i=1}^{n} \mathbb{E}_{P \sim \mathcal{Q}}\left[\mathbb{E}_{h \sim Q_i}\left[\ln \frac{Q_i(h)}{P(h)}\right]\right] \tag{13}$$

$$= D_{KL}(\mathcal{Q}||\mathcal{P}) + \sum_{i=1}^{n} \mathbb{E}_{P \sim \mathcal{Q}}\left[D_{KL}(Q_i||P)\right] \tag{14}$$

Using (11) and (14) we can bound the expected multi-task error as follows:

$$\tilde{\mathcal{L}}(\mathcal{Q}, \mathcal{D}_1, ..., \mathcal{D}_n) \leq \hat{\mathcal{L}}(\mathcal{Q}, S_1, ..., S_n) + \frac{1}{\lambda} D_{KL}(\mathcal{Q}||\mathcal{P}) + \frac{1}{\lambda} \sum_{i=1}^{n} \mathbb{E}_{P \sim \mathcal{Q}} [D_{KL}(Q_i||P)]$$

$$+ \underbrace{\frac{1}{\lambda} \ln \mathbb{E}_{P \sim \mathcal{P}} \mathbb{E}_{h \sim P} \left[ e^{\frac{\lambda}{n} \sum_{i=1}^{n} (\mathcal{L}(h, \mathcal{D}_i) - \hat{\mathcal{L}}(h, S_i))} \right]}_{\Upsilon^{\mathrm{I}}(\lambda)}$$

(15)

**Step 2 (Task environment generalization)** Now, we apply Lemma 1 on the meta-level. For that, we treat each task as random variable and instantiate the components as $X_k = \tau_i$, $l = n$ and $\mu_k = \mathcal{T}$. Furthermore, we set $\rho = \mathcal{Q}$, $\pi = \mathcal{P}$, $f = P$ and $g_k(f, X_k) = \frac{1}{n}\mathcal{L}(Q_i, \mathcal{D}_i)$. This allows us to bound the transfer error as

$$\mathcal{L}(\mathcal{Q}, \mathcal{T}) \leq \tilde{\mathcal{L}}(\mathcal{Q}, \mathcal{D}_1, ..., \mathcal{D}_n) + \frac{1}{\beta} D_{KL}(\rho||\pi) + \Upsilon^{\mathrm{II}}(\beta) \tag{16}$$

wherein $\Upsilon^{\mathrm{II}}(\beta) = \frac{1}{\beta} \ln \mathbb{E}_{P \sim \mathcal{P}} \left[ e^{\frac{\beta}{n} \sum_{i=1}^{n} \mathbb{E}_{(D,S) \sim \mathcal{T}} [\mathcal{L}(Q(P,S), \mathcal{D})] - \mathcal{L}(Q(P,S_i), \mathcal{D}_i)} \right]$.

Combining (15) with (16), we obtain

$$\mathcal{L}(\mathcal{Q}, \mathcal{T}) \leq \hat{\mathcal{L}}(\mathcal{Q}, S_1, ..., S_n) + \left( \frac{1}{\beta} + \frac{1}{\lambda} \right) D_{KL}(\mathcal{Q}||\mathcal{P})$$

$$+ \frac{1}{\lambda} \sum_{i=1}^{n} \mathbb{E}_{P \sim \mathcal{Q}} [D_{KL}(Q_i||P)] + \Upsilon^{\mathrm{I}}(\lambda) + \Upsilon^{\mathrm{II}}(\beta)$$

(17)

**Step 3 (Bounding the moment generating functions)**

$$e^{(\Upsilon^{\mathrm{I}}(\lambda) + \Upsilon^{\mathrm{II}}(\beta))} = \mathbb{E}_{P \sim \mathcal{P}} \left[ e^{\frac{\beta}{n} \sum_{i=1}^{n} \mathbb{E}_{(D,S) \sim \mathcal{T}} [\mathcal{L}(Q(P,S), \mathcal{D})] - \mathcal{L}(Q(P,S_i), \mathcal{D}_i)} \right]^{1/\beta} \cdot$$

$$\mathbb{E}_{P \sim \mathcal{P}} \mathbb{E}_{h \sim P} \left[ e^{\frac{\lambda}{n} \sum_{i=1}^{n} (\mathcal{L}(h, \mathcal{D}_i) - \hat{\mathcal{L}}(h, S_i))} \right]^{1/\lambda}$$

$$= \mathbb{E}_{P \sim \mathcal{P}} \left[ \prod_{i=1}^{n} e^{\left( \frac{\beta}{n} \mathbb{E}_{(D,S) \sim \mathcal{T}} [\mathcal{L}(Q(P,S), \mathcal{D})] \right) - \mathcal{L}(Q(P,S_i), \mathcal{D}_i)} \right]^{1/\beta} \cdot$$

(18)

$$\mathbb{E}_{P \sim \mathcal{P}} \mathbb{E}_{h \sim P} \left[ \prod_{i}^{n} \prod_{i}^{m_i} e^{\frac{\lambda}{nm_i} (\mathcal{L}(h, \mathcal{D}_i) - l(h_i, z_{ij}))} \right]^{1/\lambda}$$

**Case I: bounded loss**

If the loss function $l(h_i, z_{ij})$ is bounded in $[a_k, b_k]$, we can apply Hoeffding's lemma to each factor in (18), obtaining:

$$e^{\Upsilon^{\mathrm{I}}(\lambda) + \Upsilon^{\mathrm{II}}(\beta)} \leq \mathbb{E}_{P \sim \mathcal{P}} \left[ e^{\frac{\beta^2}{8n} (b_k - a_k)^2} \right]^{1/\beta} \cdot \mathbb{E}_{P \sim \mathcal{P}} \mathbb{E}_{h \sim P} \left[ e^{\frac{\lambda^2}{8n\tilde{m}} (b_k - a_k)^2} \right]^{1/\lambda} \tag{19}$$

$$= e^{\left( \frac{\beta}{8n} + \frac{\lambda}{8n\tilde{m}} \right) (b_k - a_k)^2} \tag{20}$$

Next, we factor out $\sqrt{n}$ from $\lambda$ and $\beta$, obtaining

$$e^{\Upsilon^{\mathrm{I}}(\lambda) + \Upsilon^{\mathrm{II}}(\beta)} = \left( e^{\Upsilon^{\mathrm{I}}(\lambda \sqrt{n}) + \Upsilon^{\mathrm{II}}(\beta \sqrt{n})} \right)^{\frac{1}{\sqrt{n}}} \tag{21}$$

Using

$$\mathbb{E}_{\mathcal{T}} \mathbb{E}_{\mathcal{D}_1} ... \mathbb{E}_{\mathcal{D}_n} \left[ e^{\Upsilon^{\mathrm{I}}(\lambda \sqrt{n}) + \Upsilon^{\mathrm{II}}(\beta \sqrt{n})} \right] \leq e^{\left( \frac{\beta}{8\sqrt{n}} + \frac{\lambda}{8\sqrt{n}\tilde{m}} \right) (b_k - a_k)^2} \tag{22}$$

we can apply Markov's inequality w.r.t. the expectations over the task distribution $\mathcal{T}$ and data distributions $\mathcal{D}_i$ to obtain that

$$\Upsilon^{\mathrm{I}}(\lambda) + \Upsilon^{\mathrm{II}}(\beta) \leq \underbrace{\frac{\beta}{8n} (b_k - a_k)^2}_{\Psi^{\mathrm{I}}(\beta)} + \underbrace{\frac{\lambda}{8n\tilde{m}} (b_k - a_k)^2}_{\Psi^{\mathrm{II}}(\lambda)} - \frac{1}{\sqrt{n}} \ln \delta \tag{23}$$

with probability at least $1 - \delta$.

**Case II: sub-gamma loss**

First, we assume that, $\forall i = 1, ..., n$ the random variables $V_i^{\mathrm{I}} := \mathcal{L}(h, \mathcal{D}_i) - l(h_i, z_{i,j})$ are *sub-gamma* with variance factor $s_{\mathrm{I}}^2$ and scale parameter $c_{\mathrm{I}}$ under the two-level prior $(\mathcal{P}, P)$ and the respective data distribution $\mathcal{D}_i$. That is, their moment generating function can be bounded by that of a Gamma distribution $\Gamma(s_{\mathrm{I}}^2, c_{\mathrm{I}})$:

$$\mathbb{E}_{z \sim \mathcal{D}_i} \mathbb{E}_{P \sim \mathcal{P}} \mathbb{E}_{h \sim P} \left[ e^{\gamma(\mathcal{L}(h, \mathcal{D}_i) - l(h, z))} \right] \leq \exp\left( \frac{\gamma^2 s_{\mathrm{I}}^2}{2(1 - c_{\mathrm{I}}\gamma)} \right) \quad \forall \gamma \in (0, 1/c_{\mathrm{I}}) \tag{24}$$

Second, we assume that, the random variable $V^{\mathrm{II}} := \mathbb{E}_{(D,S) \sim \mathcal{T}} [\mathcal{L}(Q(P,S), \mathcal{D})] - \mathcal{L}(Q(P, S_i), \mathcal{D}_i)$ is *sub-gamma* with variance factor $s_{\mathrm{II}}^2$ and scale parameter $c_{\mathrm{II}}$ under the hyper-prior $\mathcal{P}$ and the task distribution $\mathcal{T}$. That is, its moment generating function can be bounded by that of a Gamma distribution $\Gamma(s_{\mathrm{II}}^2, c_{\mathrm{II}})$:

$$\mathbb{E}_{(D,S) \sim \mathcal{T}} \mathbb{E}_{P \sim \mathcal{P}} \left[ e^{\gamma \, \mathbb{E}_{(D,S) \sim \mathcal{T}}[\mathcal{L}(Q(P,S), \mathcal{D})] - \mathcal{L}(Q(P,S), \mathcal{D})} \right] \leq \exp\left( \frac{\gamma^2 s_{\mathrm{II}}^2}{2(1 - c_{\mathrm{II}}\gamma)} \right) \quad \forall \gamma \in (0, 1/c_{\mathrm{II}}) \tag{25}$$

These two assumptions allow us to bound the expectation of (18) as follows:

$$\mathbb{E}\left[ e^{\Upsilon^{\mathrm{I}}(\lambda) + \Upsilon^{\mathrm{II}}(\beta)} \right] \leq \exp\left( \frac{\lambda s_{\mathrm{I}}^2}{2n\tilde{m}(1 - c_{\mathrm{I}}\lambda/(n\tilde{m}))} \right) \cdot \exp\left( \frac{\beta s_{\mathrm{II}}^2}{2n(1 - c_{\mathrm{II}}\beta/n)} \right) \tag{26}$$

Next, we factor out $\sqrt{n}$ from $\lambda$ and $\beta$, obtaining

$$e^{\Upsilon^{\mathrm{I}}(\lambda) + \Upsilon^{\mathrm{II}}(\beta)} = \left( e^{\Upsilon^{\mathrm{I}}(\lambda\sqrt{n}) + \Upsilon^{\mathrm{II}}(\beta\sqrt{n})} \right)^{\frac{1}{\sqrt{n}}} \tag{27}$$

Finally, by using Markov's inequality we obtain that

$$\Upsilon^{\mathrm{I}}(\lambda) + \Upsilon^{\mathrm{II}}(\beta) \leq \underbrace{\frac{\lambda s_{\mathrm{I}}^2}{2n\tilde{m}(1 - c_{\mathrm{I}}\lambda/(n\tilde{m}))}}_{\Psi^{\mathrm{I}}(\beta)} + \underbrace{\frac{\beta s_{\mathrm{II}}^2}{2n(1 - c_{\mathrm{II}}\beta/n)}}_{\Psi^{\mathrm{II}}(\lambda)} - \frac{1}{\sqrt{n}} \ln \delta \tag{28}$$

with probability at least $1 - \delta$.

To conclude the proof, we choose $\lambda = n\sqrt{\tilde{m}}$ and $\beta = \sqrt{n}$.

## A.2 Proof of Corollary 1

**Lemma 2.** *(Catoni, 2007) Let $A$ be a set, $g : A \to \mathbb{R}$ a function, and $\rho \in \mathcal{M}(A)$ and $\pi \in \mathcal{M}(A)$ probability densities over $A$. Then for any $\beta > 0$ and $\forall a \in A$,*

$$\rho^*(a) := \frac{\pi(a) e^{-\beta g(a)}}{Z} = \frac{\pi(a) e^{-\beta g(a)}}{\mathbb{E}_{a \sim \pi} \left[ e^{-\beta g(a)} \right]} \tag{29}$$

*is the minimizing probability density*

$$\underset{\rho \in \mathcal{M}(A)}{\arg\min} \ \beta \mathbb{E}_{a \sim \rho} [g(a)] + D_{KL}(\rho || \pi) . \tag{30}$$

**Proof of Corollary 1** When we choose the posterior $Q$ as the optimal Gibbs posterior $Q_i^* := Q^*(S_i, P)$, it follows that

$$\hat{\mathcal{L}}(\mathcal{Q}, S_1, ..., S_n) + \frac{1}{n} \sum_{i=1}^{n} \frac{1}{\sqrt{\tilde{m}}} \mathbb{E}_{P \sim \mathcal{Q}} \left[ D_{KL}(Q_i^* || P) \right] \tag{31}$$

$$= \frac{1}{n} \sum_{i=1}^{n} \left( \mathbb{E}_{P \sim \mathcal{Q}} \mathbb{E}_{h \sim Q_i^*} \left[ \hat{\mathcal{L}}(h, S_i) \right] + \frac{1}{\sqrt{\tilde{m}}} \left( \mathbb{E}_{P \sim \mathcal{Q}} \left[ D_{KL}(Q_i^* || P) \right] \right) \right) \tag{32}$$

$$= \frac{1}{n} \sum_{i=1}^{n} \frac{1}{\sqrt{\tilde{m}}} \left( \mathbb{E}_{P \sim \mathcal{Q}} \mathbb{E}_{h \sim Q_i^*} \left[ \sqrt{\tilde{m}} \hat{\mathcal{L}}(h, S_i) + \ln \frac{Q_i^*(h)}{P(h)} \right] \right) \tag{33}$$

$$= \frac{1}{n} \sum_{i=1}^{n} \frac{1}{\sqrt{\tilde{m}}} \left( \mathbb{E}_{P \sim \mathcal{Q}} \mathbb{E}_{h \sim Q_i^*} \left[ \frac{1}{\sqrt{\tilde{m}}} \sum_{j=1}^{m} l(h, z_i) + \ln \frac{P(h) e^{-\frac{1}{\sqrt{\tilde{m}}} \sum_{j=1}^{m} l(h, z_i)}}{P(h) Z(S_i, P)} \right] \right) \tag{34}$$

$$= \frac{1}{n} \sum_{i=1}^{n} \frac{1}{\sqrt{\tilde{m}}} \left( -\mathbb{E}_{P \sim \mathcal{Q}} \left[ \ln Z(S_i, P) \right] \right) . \tag{35}$$

This allows us to write the inequality in (3) as

$$\mathcal{L}(\mathcal{Q}, \mathcal{T}) \leq -\frac{1}{n} \sum_{i=1}^{n} \frac{1}{m_i} \mathbb{E}_{P \sim \mathcal{Q}} \left[ \ln Z(S_i, P) \right] + \left( \frac{1}{\sqrt{n}} + \frac{1}{n\sqrt{\tilde{m}}} \right) D_{KL}(\mathcal{Q} || \mathcal{P}) + C(\delta, n, \tilde{m}) . \tag{36}$$

According to Lemma 2, the Gibbs posterior $Q^*(S_i, P)$ is the minimizer of (33), in particular $\forall P \in \mathcal{M}(\mathcal{H}), \forall i = 1, ..., n$ :

$$Q^*(S_i, P) = \frac{P(h) e^{-\sqrt{\tilde{m}} \hat{\mathcal{L}}(h, S_i)}}{Z(S_i, P)} = \arg\min_{Q \in \mathcal{M}(\mathcal{H})} \mathbb{E}_{h \sim Q} \left[ \hat{\mathcal{L}}(h, S_i) \right] + \frac{1}{\sqrt{\tilde{m}}} D_{KL}(Q || P) . \tag{37}$$

Hence, we can write

$$\mathcal{L}(\mathcal{Q}, \mathcal{T}) \leq -\frac{1}{n} \sum_{i=1}^{n} \frac{1}{\sqrt{\tilde{m}}} \mathbb{E}_{P \sim \mathcal{Q}} \left[ \ln Z(S_i, P) \right] + \left( \frac{1}{\sqrt{n}} + \frac{1}{n\sqrt{\tilde{m}}} \right) D_{KL}(\mathcal{Q} || \mathcal{P}) + C(\delta, n, \tilde{m}) \tag{38}$$

$$= \frac{1}{n} \sum_{i=1}^{n} \mathbb{E}_{P \sim \mathcal{Q}} \left[ \min_{Q \in \mathcal{M}(\mathcal{H})} \hat{\mathcal{L}}(Q, S_i) + \frac{1}{\sqrt{\tilde{m}}} D_{KL}(Q || P) \right] \tag{39}$$

$$+ \left( \frac{1}{\sqrt{n}} + \frac{1}{n\sqrt{\tilde{m}}} \right) + C(\delta, n, \tilde{m}) \tag{40}$$

$$\leq \frac{1}{n} \sum_{i=1}^{n} \mathbb{E}_{P \sim \mathcal{Q}} \left[ \hat{\mathcal{L}}(Q, S_i) + \frac{1}{\sqrt{\tilde{m}}} D_{KL}(Q || P) \right] \tag{41}$$

$$+ \left( \frac{1}{\sqrt{n}} + \frac{1}{n\sqrt{\tilde{m}}} \right) D_{KL}(\mathcal{Q} || \mathcal{P}) + C(\delta, n, \tilde{m}) \tag{42}$$

$$= \hat{\mathcal{L}}(\mathcal{Q}, S_1, ..., S_n) + \left( \frac{1}{\sqrt{n}} + \frac{1}{n\sqrt{\tilde{m}}} \right) D_{KL}(\mathcal{Q} || \mathcal{P}) \tag{43}$$

$$+ \frac{1}{n} \sum_{i=1}^{n} \frac{1}{\sqrt{\tilde{m}}} \mathbb{E}_{P \sim \mathcal{Q}} \left[ D_{KL}(Q_i || P) \right] + C(\delta, n, \tilde{m}) , \tag{44}$$

which proves that the bound for Gibbs-optimal base learners in (36) and (4) is tighter than the bound in Theorem 2 which holds uniformly for all $Q \in \mathcal{M}(\mathcal{H})$.

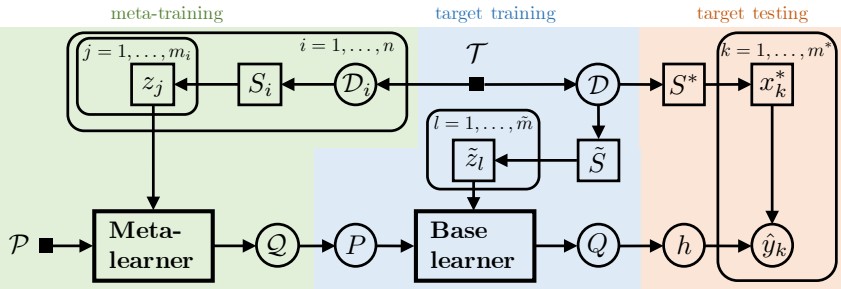

Figure S1: Overview of our meta-learning framework with environment $\mathcal{T}$, meta-task distributions $\mathcal{D}_i$, target task distribution $\mathcal{D}$, hyper-prior $\mathcal{P}$, hyper-posterior $\mathcal{Q}$, target prior $P$, target posterior $Q$, datasets $S$, and data points $z = (x, y)$.

### A.3 Proof of Proposition 1: PAC-Optimal Hyper-Posterior

An objective function corresponding to (4) reads as

$$J(\mathcal{Q}) = -\mathbb{E}_{\mathcal{Q}} \left[ \frac{1}{\sqrt{n\tilde{m}} + 1} \sum_{i=1}^{n} \ln Z(S_i, P) \right] + D_{KL}(\mathcal{Q}||\mathcal{P}) . \tag{45}$$

To obtain $J(\mathcal{Q})$, we omit all additive terms from (4) that do not depend on $\mathcal{Q}$ and multiply by the scaling factor $\frac{n\sqrt{\tilde{m}}}{\sqrt{\tilde{m}}n+1}$. Since the described transformations are monotone, the minimizing distribution of $J(\mathcal{Q})$, that is,

$$\mathcal{Q}^* = \underset{\mathcal{Q} \in \mathcal{M}(\mathcal{M}(\mathcal{H}))}{\arg\min} \ J(\mathcal{Q}) , \tag{46}$$

is also the minimizer of (4). More importantly, $J(\mathcal{Q})$ is structurally similar to the generic minimization problem in (30). Hence, we can invoke Lemma 2 with $A = \mathcal{M}(\mathcal{H})$, $g(a) = -\sum_{i=1}^{n} \ln Z(S_i, P)$, $\beta = \frac{1}{\sqrt{n\tilde{m}}+1}$, to show that the optimal hyper-posterior is

$$\mathcal{Q}^*(P) = \frac{\mathcal{P}(P) \exp\left( \frac{1}{\sqrt{n\tilde{m}}+1} \sum_{i=1}^{n} \ln Z(S_i, P) \right)}{Z^{\mathrm{II}}(S_1, ..., S_n, \mathcal{P})} , \tag{47}$$

wherein

$$Z^{\mathrm{II}}(S_1, ..., S_n, \mathcal{P}) = \mathbb{E}_{P \sim \mathcal{P}} \left[ \exp\left( \frac{1}{\sqrt{n\tilde{m}}+1} \sum_{i=1}^{n} \ln Z(S_i, P) \right) \right] .$$

$\square$

Technically, this concludes the proof of Proposition 1. However, we want to remark the following result:

If we choose $\mathcal{Q} = \mathcal{Q}^*$, the PAC-Bayes bound in (4) can be expressed in terms of the meta-level partition function $Z^{\mathrm{II}}$, that is,

$$\mathcal{L}(\mathcal{Q}, \mathcal{T}) \leq -\left( \frac{1}{\sqrt{n}} + \frac{1}{n\sqrt{\tilde{m}}} \right) \ln Z^{\mathrm{II}}(S_1, ..., S_n, \mathcal{P}) + C(\delta, n, \tilde{m}) . \tag{48}$$

We omit a detailed derivation of (48) since it is similar to the one for Corollary 1.

## B  PACOH-NN: A scalable algorithm for learning BNN priors

In this section, we summarize and further discuss our proposed meta-learning algorithm *PACOH-NN*. An overview of our proposed framework is illustrated in Figure S1. Overall, it consists of two stages *meta-training* and *meta-testing* which we explain in more details in the following.

## B.1 META-TRAINING

The hyper-posterior distribution $\mathcal{Q}$ that minimizes the upper bound on the transfer error is given by

$$\mathcal{Q}^*(P) = \frac{\mathcal{P}(P) \exp\left(\frac{1}{\sqrt{n\tilde{m}}+1} \sum_{i=1}^{n} \ln \tilde{Z}(S_i, P)\right)}{Z^{\mathrm{II}}(S_1, ..., S_n, \mathcal{P})} \tag{49}$$

Provided with a set of datasets $S_1, ..., S_2$, the meta-learner minimizes the respective meta-objective, in the case of *PACOH-SVGD*, by performing SVGD on the $\mathcal{Q}^*$. Algorithm 1 outlines the required steps in more detail.

---

**Algorithm 1** PACOH-NN-SVGD: meta-training
***
**Input:** hyper-prior $\mathcal{P}$, datasets $S_1, ..., S_n$, kernel $k(\cdot, \cdot)$, step size $\eta$, number of particles $K$
$\{\phi_1, ..., \phi_K\} \sim \mathcal{P}$                     // Initialize prior particles
**while** not converged **do**
    **for** $k = 1, ..., K$ **do**
        $\{\theta_1, ..., \theta_L\} \sim P_{\phi_k}$                // sample NN-parameters from prior
        **for** $i = 1, ..., n$ **do**
            $\ln \tilde{Z}(S_i, P_{\phi_k}) \leftarrow \mathrm{LSE}_{l=1}^{L}\left(-\sqrt{m_i}\hat{\mathcal{L}}(\theta_l, S_i)\right) - \ln L$      // estimate generalized MLL
        $\nabla_{\phi_k}\tilde{\mathcal{Q}}^*(\phi_k) \leftarrow \nabla_{\phi_k} \ln \mathcal{P}(\phi_k) + \frac{1}{\sqrt{n\tilde{m}}+1}\sum_{i=1}^{n} \nabla_{\phi_k} \ln \tilde{Z}(S_i, P_{\phi_k})$      // compute score
    $\forall k \in [K]: \phi_k \leftarrow \phi_k + \frac{\eta}{K}\sum_{k'=1}^{K}\left[k(\phi_{k'}, \phi_k)\nabla_{\phi_{k'}} \ln \tilde{\mathcal{Q}}^*(\phi_{k'}) + \nabla_{\phi_{k'}} k(\phi_{k'}, \phi_k)\right]$    // SVGD
**Output:** set of priors $\{P_{\phi_1}, ..., P_{\phi_K}\}$

---

Alternatively, to estimate the score of $\nabla_{\phi_k}\tilde{\mathcal{Q}}^*(\phi_k)$ we can use mini-batching at both the task and the dataset level. Specifically, for a given meta-batch size of $n_{bs}$ and a batch size of $m_{bs}$, we get Algorithm 2.

---

**Algorithm 2** PACOH-NN-SVGD: mini-batched meta-training
***
**Input:** hyper-prior $\mathcal{P}$, datasets $S_1, ..., S_n$
**Input:** kernel function $k(\cdot, \cdot)$, SVGD step size $\eta$, number of particles $K$
$\{\phi_1, ..., \phi_K\} \sim \mathcal{P}$                     // Initialize prior particles
**while** not converged **do**
    $\{T_1, ..., T_{n_{bs}}\} \subseteq [n]$            // sample $n_{bs}$ tasks uniformly at random
    **for** $i = 1, ..., n_{bs}$ **do**
        $\tilde{S}_i \leftarrow \{z_1, ..., z_{m_{bs}}\} \subseteq S_{T_i}$      // sample $m_{bs}$ datapoints from $S_{T_i}$ uniformly at random
    **for** $k = 1, ..., K$ **do**
        $\{\theta_1, ..., \theta_L\} \sim P_{\phi_k}$               // sample NN-parameters from prior
        **for** $i = 1, ..., n_{bs}$ **do**
            $\ln \tilde{Z}(\tilde{S}_i, P_{\phi_k}) \leftarrow \mathrm{LSE}_{l=1}^{L}\left(-\sqrt{m_i}\hat{\mathcal{L}}(\theta_l, \tilde{S}_i)\right) - \ln L$      // estimate generalized MLL
        $\nabla_{\phi_k}\tilde{\mathcal{Q}}^*(\phi_k) \leftarrow \nabla_{\phi_k} \ln \mathcal{P}(\phi_k) + \frac{1}{\sqrt{n\tilde{m}}+1}\frac{n}{n_{bs}}\sum_{i=1}^{n_{bs}} \nabla_{\phi_k} \ln \tilde{Z}(S_i, P_{\phi_k})$    // compute score
    $\phi_k \leftarrow \phi_k + \frac{\eta}{K}\sum_{k'=1}^{K}\left[k(\phi_{k'}, \phi_k)\nabla_{\phi_{k'}} \ln \tilde{\mathcal{Q}}^*(\phi_{k'}) + \nabla_{\phi_{k'}} k(\phi_{k'}, \phi_k)\right] \forall k \in [K]$    // update
part.
**Output:** set of priors $\{P_{\phi_1}, ..., P_{\phi_K}\}$

---

## B.2 META-TESTING

The meta-learned prior knowledge is now deployed by a base learner. The base learner is given a training dataset $\tilde{S} \sim \mathcal{D}$ pertaining to an unseen task $\tau = (\mathcal{D}, m) \sim \mathcal{T}$. With the purpose of approximating the generalized Bayesian posterior $Q^*(S, P)$, the base learner performs (normal) posterior inference. Algorithm 3 details the steps of the approximating procedure – referred to as *target training* – when performed via SVGD. For a data point $x^*$, the respective predictor outputs a

probability distribution given as $\tilde{p}(y^*|x^*, \tilde{S}) \leftarrow \frac{1}{K \cdot L} \sum_{k=1}^{K} \sum_{l=1}^{L} p(y^*|h_{\theta_l^k}(x^*))$. We evaluate the quality of the predictions on a held-out test dataset $\tilde{S}^* \sim \mathcal{D}$ from the same task, in a *target testing phase* (see Appendix C.2).

---

**Algorithm 3** PACOH-NN-SVGD: meta-testing

---

**Input:** set of priors $\{P_{\phi_1}, ..., P_{\phi_K}\}$, target training dataset $\tilde{S}$, evaluation point $x^*$
**Input:** kernel function $k(\cdot, \cdot)$, SVGD step size $\nu$, number of particles $L$
**for** $k = 1, ..., K$ **do**
    $\{\theta_1^k, ..., \theta_L^k\} \sim P_{\phi_k}$                          // initialize NN posterior particles from $k$-th prior
    **while** not converged **do**
        **for** $l = 1, ..., L$ **do**
            $\nabla_{\theta_l^k} Q^*(\theta_l^k)) \leftarrow \nabla_{\theta_l^k} \ln P_{\phi_k}(\theta_l^k)) + \sqrt{m} \, \nabla_{\theta_l^k} \mathcal{L}(l, \tilde{S})$          // compute score
        $\theta_l^k \leftarrow \theta_l^k + \frac{\nu}{L} \sum_{l'=1}^{L} \left[ k(\theta_{l'}^k, \theta_l^k) \nabla_{\theta_{l'}^k} \ln Q^*(\theta_{l'}^k) + \nabla_{\theta_{l'}^k} k(\theta_{l'}^k, \theta_l^k) \right] \forall l \in [L]$      // update
particles
**Output:** a set of NN parameters $\bigcup_{k=1}^{K} \{\theta_1^k ..., \theta_L^k\}$

---

## B.3 PROPERTIES OF THE SCORE ESTIMATOR

Since the marginal log-likelihood of BNNs is intractable, we have replaced it by a numerically stable Monte Carlo estimator $\ln \tilde{Z}(S_i, P_\phi)$ in (7), in particular

$$\ln \tilde{Z}(S_i, P_\phi) := \ln \frac{1}{L} \sum_{l=1}^{L} e^{-\sqrt{m_i} \hat{\mathcal{L}}(\theta_l, S_i)} = \text{LSE}_{l=1}^{L} \left( -\sqrt{m_i} \hat{\mathcal{L}}(\theta_l, S_i) \right) - \ln L , \ \ \theta_l \sim P_\phi . \quad (50)$$

Since the Monte Carlo estimator involves approximating an expectation of an exponential, it is not unbiased. However, we can show that replacing $\ln Z(S_i, P_\phi)$ by the estimator $\ln \tilde{Z}(S_i, P_\phi)$, we still minimize a valid upper bound on the transfer error (see Proposition 3).

**Proposition 3.** *In expectation, replacing* $\ln Z(S_i, P_\phi)$ *in (4) by the Monte Carlo estimate* $\ln \tilde{Z}(S_i, P) := \ln \frac{1}{L} \sum_{l=1}^{L} e^{-\sqrt{m_i} \hat{\mathcal{L}}(\theta_l, S_i)}$, $\theta_l \sim P$ *still yields an valid upper bound of the transfer error. In particular, it holds that*

$$\mathcal{L}(\mathcal{Q}, \mathcal{T}) \leq -\frac{1}{n} \sum_{i=1}^{n} \frac{1}{\sqrt{\tilde{m}}} \mathbb{E}_{P \sim \mathcal{Q}} \left[ \ln Z(S_i, P) \right] + \left( \frac{1}{\sqrt{n}} + \frac{1}{n\sqrt{\tilde{m}}} \right) D_{KL}(\mathcal{Q}||\mathcal{P}) + C(\delta, n, \tilde{m}) \ (51)$$

$$\leq -\frac{1}{n} \sum_{i=1}^{n} \frac{1}{\sqrt{\tilde{m}}} \mathbb{E}_{P \sim \mathcal{Q}} \left[ \mathbb{E}_{\theta_1, ..., \theta_L \sim P} \left[ \ln \tilde{Z}(S_i, P) \right] \right]$$
$$+ \left( \frac{1}{\sqrt{n}} + \frac{1}{n\sqrt{\tilde{m}}} \right) D_{KL}(\mathcal{Q}||\mathcal{P}) + C(\delta, n, \tilde{m}). \quad (52)$$

*Proof.* Firsts, we show that:

$$\mathbb{E}_{\theta_1, ..., \theta_L \sim P} \left[ \ln \tilde{Z}(S_i, P) \right] = \mathbb{E}_{\theta_1, ..., \theta_L \sim P} \left[ \ln \frac{1}{L} \sum_{l=1}^{L} e^{-\sqrt{m_i} \hat{\mathcal{L}}(\theta_l, S_i)} \right]$$

$$\leq \ln \frac{1}{L} \sum_{l=1}^{L} \mathbb{E}_{\theta_l \sim P} \left[ e^{-\sqrt{m_i} \hat{\mathcal{L}}(\theta_l, S_i)} \right]$$

$$= \ln \mathbb{E}_{\theta \sim P} \left[ e^{-\sqrt{m_i} \hat{\mathcal{L}}(\theta, S_i)} \right]$$

$$= \ln Z(S_i, P) \quad (53)$$

which follows directly from Jensen's inequality and the concavity of the logarithm. Now, Proposition 3 follows directly from (53). $\qquad \square$

In fact, by the law of large numbers, it is straightforward to show that as $L \to \infty$, the $\ln \tilde{Z}(S_i, P) \xrightarrow{\text{a.s.}} \ln Z(S_i, P)$, that is, the estimator becomes asymptotically unbiased and we recover the original PAC-Bayesian bound (i.e. (52) $\xrightarrow{\text{a.s.}}$ (51)). Also it is noteworthy that the bound in (52) we get by our estimator is, in expectation, tighter than the upper bound when using the naive estimator

$$\ln \hat{Z}(S_i, P) := -\sqrt{m_i} \frac{1}{L} \sum_{l=1}^{L} \hat{\mathcal{L}}(\theta_l, S_i) \quad \theta_l \sim P_\phi$$

which can be obtained by applying Jensen's inequality to $\ln \mathbb{E}_{\theta \sim P_\phi} \left[ e^{-\sqrt{m_i} \hat{\mathcal{L}}(\theta, S_i)} \right]$. In the edge case $L = 1$ our LSE estimator $\ln \tilde{Z}(S_i, P)$ falls back to this naive estimator and coincides in expectation with $\mathbb{E}[\ln \hat{Z}(S_i, P)] = -\sqrt{m_i} \, \mathbb{E}_{\theta \sim P} \hat{\mathcal{L}}(\theta_l, S_i)$. As a result, we effectively minimize the looser upper bound

$$\mathcal{L}(\mathcal{Q}, \mathcal{T}) \leq \frac{1}{n} \sum_{i=1}^{n} \mathbb{E}_{\theta \sim P} \left[ \hat{\mathcal{L}}(\theta, S_i) \right] + \left( \frac{1}{\sqrt{n}} + \frac{1}{n\sqrt{\tilde{m}}} \right) D_{KL}(\mathcal{Q}||\mathcal{P}) + C(\delta, n, \tilde{m}). \tag{54}$$

$$= \mathbb{E}_{\theta \sim P} \left[ \frac{1}{n\tilde{m}} \sum_{i=1}^{n} \sum_{j=1}^{m_i} - \ln p(y_{ij}|x_{ij}, \theta) \right] + \left( \frac{1}{\sqrt{n}} + \frac{1}{n\sqrt{\tilde{m}}} \right) D_{KL}(\mathcal{Q}||\mathcal{P}) + C(\delta, n, \tilde{m}) \tag{55}$$

As we can see from (55), the boundaries between the tasks vanish in the edge case of $L = 1$, that is, all data-points are treated as if they would belong to one dataset. This suggests that $L$ should be chosen greater than one. In our experiments, we used $L = 5$ and found the corresponding approximation to be sufficient.

## C   EXPERIMENTS

### C.1   META-LEARNING ENVIRONMENTS

In this section, we provide further details on the meta-learning environments used in Section 6. Information about the numbers of tasks and samples in the respective environments can be found in Table S1.

|       | Sinusoid | Cauchy | SwissFEL | Physionet | Berkeley |
|-------|----------|--------|----------|-----------|----------|
| $n$   | 20       | 20     | 5        | 100       | 36       |
| $m_i$ | 5        | 20     | 200      | 4 - 24    | 288      |

Table S1: Number of tasks $n$ and samples per task $m_i$ for the different meta-learning environments.

### C.1.1   SINUSOIDS

Each task of the sinusoid environment corresponds to a parametric function

$$f_{a,b,c,\beta}(x) = \beta * x + a * \sin(1.5 * (x - b)) + c \,, \tag{56}$$

which, in essence, consists of an affine as well as a sinusoid function. Tasks differ in the function parameters $(a, b, c, \beta)$ that are sampled from the task environment $\mathcal{T}$ as follows:

$$a \sim \mathcal{U}(0.7, 1.3), \quad b \sim \mathcal{N}(0, 0.1^2), \quad c \sim \mathcal{N}(5.0, 0.1^2), \quad \beta \sim \mathcal{N}(0.5, 0.2^2) \,. \tag{57}$$

Figure S2a depicts functions $f_{a,b,c,\beta}$ with parameters sampled according to (57). To draw training samples from each task, we draw $x$ uniformly from $\mathcal{U}(-5, 5)$ and add Gaussian noise with standard deviation 0.1 to the function values $f(x)$:

$$x \sim \mathcal{U}(-5, 5) \,, \qquad y \sim \mathcal{N}(f_{a,b,c,\beta}(x), 0.1^2) \,. \tag{58}$$

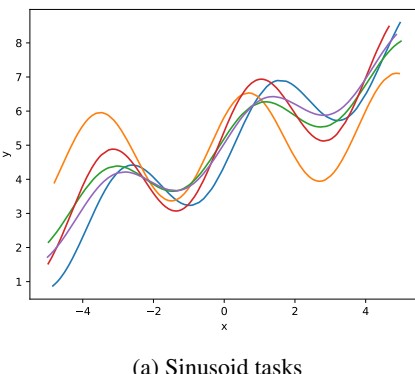
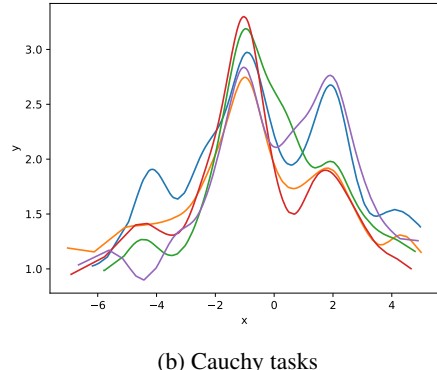

(a) Sinusoid tasks                    (b) Cauchy tasks

Figure S2: Depiction of tasks (i.e., functions) sampled from the Sinusoid and Cauchy task environment, respectively. Note that the Cauchy task environment is two-dimensional ($\dim(\mathcal{X}) = 2$), while (b) displays a one-dimensional projection.

### C.1.2 CAUCHY

Each task of the Cauchy environment can be interpreted as a two dimensional mixture of Cauchy distributions plus a function sampled from a Gaussian process prior with zero mean and SE kernel function $k(x, x') = \exp\left(\frac{||x - x'||_2^2}{2l}\right)$ with $l = 0.2$. The (unnormalized) mixture of Cauchy densities is defined as:

$$m(x) = \frac{6}{\pi \cdot (1 + ||x - \mu_1||_2^2)} + \frac{3}{\pi \cdot (1 + ||x - \mu_2||_2^2)} \,, \tag{59}$$

with $\mu_1 = (-1, -1)^\top$ and $\mu_2 = (2, 2)^\top$.

Functions from the task environments are sampled as follows:

$$f(x) = m(x) + g(x) \,, \qquad g \sim \mathcal{GP}(0, k(x, x')) \,. \tag{60}$$

Figure S2b depicts a one-dimensional projection of functions sampled according to (60). To draw training samples from each task, we draw $x$ from a truncated normal distribution and add Gaussian noise with standard deviation $0.05$ to the function values $f(x)$:

$$x := \min\{\max\{\tilde{x}, 2\}, -3\} \,, \quad \tilde{x} \sim \mathcal{N}(0, 2.5^2) \,, \qquad y \sim \mathcal{N}(f(x), 0.05^2) \,. \tag{61}$$

### C.1.3 SWISSFEL

Free-electron lasers (FELs) accelerate electrons to very high speed in order to generate shortly pulsed laser beams with wavelengths in the X-ray spectrum. These X-ray pulses can be used to map nanometer scale structures, thus facilitating experiments in molecular biology and material science. The accelerator and the electron beam line of a FEL consist of multiple magnets and other adjustable components, each of which has several parameters that experts adjust to maximize the pulse energy (Kirschner et al., 2019a). Due do different operational modes, parameter drift, and changing (latent) conditions, the laser's pulse energy function, in response to its parameters, changes across time. As a result, optimizing the laser's parameters is a recurrent task.

Overall, our meta-learning environment consists of different parameter optimization runs (i.e., tasks) on the SwissFEL, an 800 meter long laser located in Switzerland (Milne et al., 2017). A picture of the SwissFEL is shown in Figure S3. The input space, corresponding to the laser's parameters, has 12 dimensions whereas the regression target is the pulse energy (1-dimensional). For details on the individual parameters, we refer to Kirschner et al. (2019b). For each run, we have around 2000 data points. Since these data-points are generated with online optimization methods, the data are non-i.i.d. and get successively less diverse throughout the optimization. As we are concerned with meta-learning with limited data and want to avoid issues with highly dependent data points, we only take the first 400 data points per run and split them into training and test subsets of size 200. Overall,

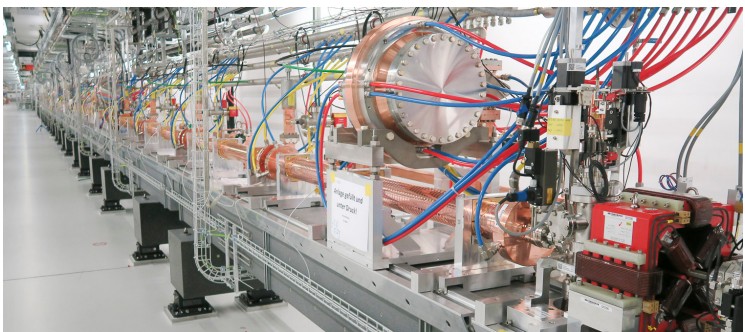

Figure S3: Accelerator of the Swiss Free-Electron Laser (SwissFEL).

we have 9 runs (tasks) available. 5 of those runs are used for meta-training and the remaining 4 runs are used for meta-testing.

### C.1.4 PHYSIONET

The 2012 Physionet competition (Silva et al., 2012) published an open-access dataset of patient stays on the intensive care unit (ICU). Each patient stay consists of a time series over 48 hours, where up to 37 clinical variables are measured. The original task in the competition was binary classification of patient mortality, but due to the large number of missing values (around 80 % across all features), the dataset is also popular as a test bed for time series prediction methods, especially using Gaussian processes (Fortuin et al., 2019).

In this work, we treat each patient as a separate task and the different clinical variables as different environments. We use the Glasgow coma scale (GCS) and hematocrit value (HCT) as environments for our study, since they are among the most frequently measured variables in this dataset. From the dataset, we remove all patients where less than four measurements of CGS (and HCT respectively) are available. From the remaining patients we use 100 patients for meta-training and 500 patients each for meta-validation and meta-testing. Here, each patient corresponds to a task. Since the number of available measurements differs across patients, the number of training points $m_i$ ranges between 4 and 24.

### C.1.5 BERKELEY-SENSOR

We use data from 46 sensors deployed in different locations at the Intel Research lab in Berkeley (Madden, 2004). The dataset contains 4 days of data, sampled at 10 minute intervals. Each task corresponds to one of the 46 sensors and requires auto-regressive prediction, in particular, predicting the next temperature measurement given the last 10 measurement values. In that, 36 sensors (tasks) with data for the first two days are use for meta-training and whereas the remaining 10 sensors with data for the last two days are employed for meta-testing. Note, that we separate meta-training and -testing data both temporally and spatially since the data is non-i.i.d. For the meta-testing, we use the 3rd day as context data, i.e. for target training and the remaining data for target testing.

### C.2 EXPERIMENTAL METHODOLOGY

In the following, we describe our experimental methodology and provide details on how the empirical results reported in Section 6 were generated. Overall, evaluating a meta-learner consists of two phases, *meta-training* and *meta-testing*, outlined in Appendix B. The latter can be further sub-divided into *target training* and *target testing*. Figure S1 illustrates these different stages for our PAC-Bayesian meta-learning framework.

The outcome of the training procedure is an approximation for the generalized Bayesian posterior $Q^*(S, P)$ (see Appendix ), pertaining to an unseen task $\tau = (\mathcal{D}, m) \sim \mathcal{T}$ from which we observe a dataset $\tilde{S} \sim \mathcal{D}$. In *target-testing*, we evaluate its predictions on a held-out test dataset $\tilde{S}^* \sim \mathcal{D}$ from the same task. For PACOH-NN, NPs and MLAP the respective predictor outputs a probability distribution $\hat{p}(y^*|x^*, \tilde{S})$ for the $x^*$ in $\tilde{S}^*$. The respective mean prediction corresponds to the expectation

of $\hat{p}$, that is $\hat{y} = \hat{\mathbb{E}}(y^*|x^*, \tilde{S})$. In the case of MAML, only a mean prediction is available. Based on the mean predictions, we compute the *root mean-squared error (RMSE)*:

$$\text{RMSE} = \sqrt{\frac{1}{|\tilde{S}^*|} \sum_{(x^*, y^*) \in S^*} (y^* - \hat{y})^2} \ . \tag{62}$$

and the *calibration error* (see Appendix C.2.1). Note that unlike e.g. Rothfuss et al. (2019a) who report the test log-likelihood, we aim to measure the quality of mean predictions and the quality of uncertainty estimate separately, thus reporting both RMSE and calibration error.

The described meta-training and meta-testing procedure is repeated for five random seeds that influence both the initialization and gradient-estimates of the concerned algorithms. The reported averages and standard deviations are based on the results obtained for different seeds.

### C.2.1 CALIBRATION ERROR

The concept of calibration applies to probabilistic predictors that, given a new target input $x_i$, produce a probability distribution $\hat{p}(y_i|x_i)$ over predicted target values $y_i$. Corresponding to the predictive density, we denote a predictor's cumulative density function (CDF) as $\hat{F}(y_j|x_j) = \int_{-\infty}^{y_j} \hat{p}(y|x_i)dy$. For confidence levels $0 \le q_h < ... < q_H \le 1$, we can compute the corresponding empirical frequency

$$\hat{q}_h = \frac{|\{y_j \mid \hat{F}(y_j|x_j) \le q_h, j = 1, ..., m\}|}{m} \ , \tag{63}$$

based on dataset $S = \{(x_i, y_i)\}_{i=1}^m$ of $m$ samples. If we have calibrated predictions we would expect that $\hat{q}_h \to q_h$ as $m \to \infty$. Similar to (Kuleshov et al., 2018), we can define the calibration error as a function of residuals $\hat{q}_h - q_h$, in particular,

$$\text{calib-err} = \frac{1}{H} \sum_{h=1}^{H} |\hat{q}_h - q_h| \ . \tag{64}$$

Note that we while (Kuleshov et al., 2018) reports the average of squared residuals $|\hat{q}_h - q_h|^2$, we report the average of absolute residuals $|\hat{q}_h - q_h|$ in order to preserve the units and keep the calibration error easier to interpret. In our experiments, we compute (64) with $K = 20$ equally spaced confidence levels between 0 and 1.

### C.3 HYPER-PARAMETER SELECTION

For each of the meta-environments and algorithms, we ran a separate hyper-parameter search to select the hyper-parameters. In particular, we use the `hyperopt`[1] package (Bergstra et al., 2013) which performs Bayesian optimization based on regression trees. As optimization metric, we employ the average log-likelihood, evaluated on a separate validation set of tasks.

The scripts for reproducing the hyper-parameter search are included in our code repository[2] For the reported results, we provide the selected hyper-parameters and detailed evaluation results under `[Link will be made added upon acceptance]`

### C.4 META-LEARNING FOR BANDITS - VACCINE DEVELOPMENT

In this section, we provide additional details on the experiment in Section 6.2.

We use data from Widmer et al. (2010) which contains the binding affinities ($IC_{50}$ values) of many peptide candidates to seven different MHC-I alleles. Peptides with $IC_{50} > 500nM$ are considered non-binders, all others binders. Following Krause & Ong (2011), we convert the $IC_{50}$ values into negative log-scale and normalize them such that 500nM corresponds to zero, i.e. $r := -\log_{10}(IC_{50}) + \log_{10}(500)$ with is used as reward signal of our bandit.

---

[1] http://hyperopt.github.io/hyperopt/
[2] [Link will be made added upon acceptance]

| Allele | A-0202 | A-0203 | A-0201 | A-2301 | A-2402 |
|--------|--------|--------|--------|--------|--------|
| $m_i$ | 1446 | 1442 | 3088 | 103 | 196 |

Table S2: MHC-I alleles used for meta-training and their corresponding number of meta-training samples $m_i$.

We use 5 alleles to meta-learn a BNN prior. The alleles and the corresponding number of data points, available for meta-training, are listed in Table S2. The most genetically dissimilar allele (A-6901) is used for our bandit task. In each iteration, the experimenter (i.e. bandit algorithm) chooses to test one peptide among the pool of 813 candidates and receives $r$ as a reward feedback. Hence, we are concerned with a 813-arm bandit wherein the action $a_t \in \{1, ..., 813\} = \mathcal{A}$ in iteration $t$ corresponds to testing $a_t$-th peptide candidate. In response, the algorithm receives the respective negative log-IC$_{50}$ as reward $r(a_t)$.

As metrics, we report the *average regret*

$$R_T^{avg.} := \max_{a \in \mathcal{A}} r(a) - \frac{1}{T} \sum_{t=1}^{T} r(a_t)$$

and the *simple regret*

$$R_T^{simple} := \max_{a \in \mathcal{A}} r(a) - \max_{t=1,...,T} r(a_t)$$

To ensure a fair comparison, the prior parameters of the GP for GP-UCB and GP-TS are meta-learned by minimizing the GP's marginal log-likelihood on the five meta-training tasks. For the prior, we use a constant mean function and tried various kernel functions (linear, SE, Matern). Due to the 45-dimensional feature space, we found the linear kernel to work the best. So overall, the constant mean and the variance parameter of the linear kernel are meta-learned.

