# OpenReview forum: "Meta-Learning Bayesian Neural Network Priors Based on PAC-Bayesian Theory"
_ICLR.cc/2021/Conference — Reject_

### Official Review · AnonReviewer2 · 2020-10-27
**PAC-Bayesian meta learning with Bayesian neural nets**

**Rating:** 4
**Confidence:** 4

**Review:**

This submission has significant overlap with the following reference:

Jonas Rothfuss, Vincent Fortuin, and Andreas Krause. PACOH: Bayes-Optimal Meta-Learning with
PAC-Guarantees. arXiv, 2020.

Assumably, the submission is from the same authors as theoretical analyses and experiment setups are almost identical. The main difference I can see is the Bayesian baseline model is Bayesian neural nets while the reference used Gaussian processes. The rating I gave is because of this main concern on overlapping content.

The analysis of PAC-Bayesian bounds led to the conclusion that Gibbs posterior can help minimize the bounds in the meta-learning settings. With Gibbs base learner, the bound expressed with the corresponding partition function can be done more efficiently compared to existing PAC-Bayesian meta-learners, including MLAP.

My other concerns are as follows:

1. The authors may need to make sure about the main contributions of this submission with most of theoretical results already presented in the referred paper. If Bayesian neural nets are indeed showing much better empirical performances compared to GPs, the authors may need to emphasize that and discuss thoroughly with performance comparison.

2. The final solution to derive Gibbs posterior is just deriving the model posterior given the hyper-prior using the data from all training tasks. If that is the case, the authors may present it clearly and discuss the connections as well as pros and cons to other existing meta-learners, for example, the existing metric-based methods that reweighting the loss across training tasks.

3. The experiments may need to be done more consistently. For example, it is not clear why figure 1 only compares PACOHNN with a vanilla BNN using a simply prior instead of, for example, comparing with MLAP or PACOH with GPs? For classification examples, the authors did not compare with neural processes as claimed at the beginning of section 6.1. In section 6.2, why max regrets are smaller than average regrets in figure 3? How many training samples were used for the five tasks to get the reported results?

4. The writing of theoretical analyses (main text, but especially appendix) can be improved. There are typos and confusing math notations. For example, in section 5,  above equation 7, "\epsilon_l j ", where j appears to be a typo? In A.1, "to proof the theorem" should be "to prove the theorem..."; in step 1 and 2 of the proof, X_k was used to denote the data point and then task, respectively, which is indeed confusing. It is not clear either what the subscript k was used for. Cleaner math notations, better organization, and careful rewriting may significantly help the readability of the paper.

5. The authors may want to add complexity analysis of PACOH-NN. It is in fact quite confusing to me why it "maintains a nearly constant memory and compute load as the number of tasks grow." as stated since the number of tasks does factor in based on Algorithm 1 in Appendix B. Clearer setup descriptions with complexity analysis and discussion are needed.

---

> ### Author Response · Authors · 2020-11-22
> **Response to AnonReviewer2**
>
> Dear reviewer,
>
> Thank you for your detailed feedback. In what follows, we will address your concerns in order:
>
> ---
> 1.
> While our approach indeed builds on the methodology developed in Rothfuss et al. (2020), the present submission goes substantially beyond it, as elaborated below. Crucially, in terms of algorithms, previous work was restricted to tractable base learners (e.g. GPs).  We theoretically show how to lift these restrictions, and present extensive empirical demonstrations on (intractable) Bayesian neural network base learners.
>
> To ensure our paper is self-contained, Section 4 indeed has some overlap as we build on the methodology of Rothfuss et al. (2020). It provides necessary background to understand the PAC-Bayesian setup and how to arrive at the PAC-optimal hyper-posterior. Would you prefer if we remove large parts of Section 4 and just state the end-result, e.g., Proposition 1?
>
> Moreover, we want to emphasize that, thanks to our improved proof methodology, our bounds are tighter than the ones in Rothfuss et al. (2020). We extended the corresponding discussion under Theorem 2 in the reviser paper to point this out more clearly.
>
> We acknowledge that in the submitted version of the paper, the PACOH-NN algorithm is presented only very quickly and is discussed insufficiently. Aiming to address this, we have significantly extended Section 5 and Appendix B. In particular, we have added
> *   a detailed discussion of the properties / approximation quality of our proposed generalized marginal log-likelihood estimator
> *   a description of the proposed PACOH-NN algorithm
> *   a discussion of PACOH-NN’s advantages over previous methods
> to the revised paper.
>
> Overall, the key algorithmic contribution of our paper over Rothfuss et al. (2020) is the numerically stable score estimator for the generalized marginal log-likelihood (MLL) and our analysis which shows that despite the approximation error, we still minimize a valid upper bound on the transfer error. This makes PAC-Bayesian meta-learning both tractable and scalable for a much larger array of models, including BNNs. In contrast, Rothfuss et al. is restricted to models with closed-form MLL which severely restricts its use. PACOH-NN can also be used for tasks such as image classification or high-dimensional Bayesian optimization (c.f. Sec. 6.2.) to which the method of Rothfuss et al. (2020) is not applicable. Hence, we rather consider PACOH-NN complementary to the GP based method of Rothfuss et al. (2020) than as a replacement.
>
> Overall, we believe that making PAC-Bayesian meta-learning scale to large-scale settings and work on challenging real-world problems is a relevant contribution and certainly of interest to the ICLR community.
>
> ---
> 2.
>  Importantly, the (Gibbs) hyper-posterior in Prop. 1 is not equivalent to the posterior given hyper-prior and likelihood of the union of all datasets. For instance, this can be shown by the convexity of the log-Laplace transform in $\ln Z(S_i, P_\phi)$. In fact, the log-Laplace transform makes sure that the boundaries between tasks are implicitly taken into account which is crucial if we want to successfully meta-learn. Only in the edge case $L=1$ of our (generalized) marginal log-likelihood estimator, the log-sum-exp is no longer strictly convex and becomes the (linear) identity function. In this edge case, we effectively fall back on the likelihood of the union of all datasets which ignores the task boundaries. This is discussed in Sec. 5 and Appendix B of the revised paper.
>
> ---
>
> 3.
> Figure 1 is meant as a brief qualitative inspection that compares a standard Gaussian BNN prior with a meta-learned one. For the purpose of benchmarking against other meta-learners, we provide extensive quantitative results in Table 1, 2 and 3.
>
> We did not find a neural process implementation for classification. The authors only provide one for regression. If you can point us to a NP for classification implementation, we are happy to include it into our experiments.
>
> We agree that the wording “maximum regret” is misleading and renamed it into “simple regret”, referring to $\max_{a} r(a) - \max_{t=1,...,T}r(a_t)$. We added details on the bandit experiment and regret definitions to Sec. C.4 of the revised paper. Table S2 reports the number of samples per tasks (103 - 3088), which substantially varies across the 5 alleles.
>
> ---
>  4.
>  We have fixed typos and inconsistencies in the mathematical notation.
>
> ---
>
> 5.
>  Thanks for noticing the ambiguity in the complexity analysis. To address this issue, we have added a memory and computation complexity analysis of the algorithm to Section 5 and added the pseudocode for a batched version of Algorithm 1 in Appendix B. Now the computational properties of PACOH-NN are discussed in detail and compared to previous methods such as MLAP.
> ---
>
> Finally, we want to thank you once again for the provided feedback, which allowed us to provide a better emphasis on the strengths and relevance of our approach.

---

> > ### Comment · AnonReviewer2 · 2020-11-23
> > **a couple of remaining questions**
> >
> > This reviewer truly appreciates the detailed response from the authors, especially clarification of the new developments compared to Rothfuss et al. (2020).
> >
> > Regarding "Would you prefer if we remove large parts of Section 4 and just state the end-result, e.g., Proposition 1?"
> >
> > I believe it is fine to have additional content in addition to the main results but I do appreciate the efforts if the authors can clearly distinguish the new results from the previous efforts.
> >
> > Also, is there a specific reason the authors did not compare PACOH-NN with the GP-based method in Rothfuss et al. (2020)? It would definitely help better clarify the significance of this paper compared to Rothfuss et al. (2020) if the authors can do so.
> >
> >
> >
> >
> > Regarding the question on neural process, in Section 6.1 (page 7) of the current revision, it still reads "Second, we compare our proposed approach against various popular meta-learning algorithms, including neural processes (NPs) ..." If there is no straightforward to implement it for classification, the authors may need to revise accordingly.

---

> > > ### Author Response · Authors · 2020-11-25
> > > **answer to the remaining questions**
> > >
> > > > I believe it is fine to have additional content in addition to the main results but I do appreciate the efforts if the authors can clearly distinguish the new results from the previous efforts.
> > >
> > > In the revised version of the paper, especially in Section 4, we try to stress out more clearly when we rely on the methodology of previous work and when new results are contributed. We hope that this resolves your concerns.
> > >
> > > > Also, is there a specific reason the authors did not compare PACOH-NN with the GP-based method in Rothfuss et al. (2020)? It would definitely help better clarify the significance of this paper compared to Rothfuss et al. (2020) if the authors can do so.
> > >
> > > Originally we did not include the GP-based method of Rothfuss et al. (2020) in the benchmark experiments due to space constraints and with the reasoning to restrict the comparison to NN-based methods so that the performance differences reflect upon the meta-learning algorithm rather than on the prediction model. However, we agree with the reviewer that a comparison to Rothfuss et al. (2020) would be helpful. Hence, the last two days, we ran experiments with PACOH-GP (Rothfuss et al., 2020) under similar conditions like PACOH-NN (e.g. $K=5$ SVGD particles) and added them to Table 1 and 2 of the revised paper. We hope that this helps towards further clarifying the significance of the paper.
> > >
> > > > Regarding the question on neural process, in Section 6.1 (page 7) of the current revision, it still reads "Second, we compare our proposed approach against various popular meta-learning algorithms, including neural processes (NPs) ..." If there is no straightforward to implement it for classification, the authors may need to revise accordingly.
> > >
> > > Thanks for pointing that out. We have revised the respective paragraph in Section 6.1 accordingly.

---

### Official Review · AnonReviewer1 · 2020-10-27
**Review of the PACOH-NN algorithm for learning BNN priors**

**Rating:** 7
**Confidence:** 2

**Review:**

### Summary:

One of the main issues that BNNs have to face is the choice of good informative priors in order to provide precise information about the uncertainty of predictions. The present work connects BNNs and PAC theory to formulate a new system to obtain a general-purpose approach for obtaining significative priors for Bayesian NNs. This is done by employing the closed-form solution for the PAC-Bayesian meta-learning problem. The meta learner here learns a hyper-posterior over the priors of the parameters of the BNN by using the closed-form expression PACOH  (PAC Optimal Hyper-Posterior). This is applied in the context of NNs, where the priors are to be defined over the NN parameters. Extensive experiments are carried out to show the performance both in regression and classification datasets, as well as its scalability. In all of these regards, the system seems to be competitive, improving on the results of previous state-of-the-art methods and producing promising results in real-world problems.

##### Pros:

* The method does not employ nested optimization problems, therefore avoiding all the issues related to these approaches altogether.
* The usage of Bayesian seems to point in the right direction since the Bayesian framework allows for an easy formulation of the different levels needed here to formulate the system.
* The construction of the PACOH closed-form expression seems innovative and relevant
* The final system improves on the previous state-of-the-art in most of the cases here shown, and in some experiments, the improvement is very clear. Thanks to the fact that it is agnostic to the inference method in use, it presents itself as a very general-purposed approach, able to improve the predictive qualities of previous methods.
* The article is very complete and detailed, both in the main body and in the appendix. The appendix is particularly extensive, providing insight on many of the main points of the main text.
* The experiments conducted are exhaustive and complete, providing a very wide scope of the capabilities of the method. Detailed results can be found for every experiment and task.
* The text is well written and comprehensible

##### Cons:

* The choice in section 5 of using Gaussian priors with diagonal covariance matrices is motivated by the convenience in the computations. Moreover, the hyper-prior is also modeled by a zero-centered spherical Gaussian. How does choosing these distributions affect the final results? Are there been any experiments on which the parametric distributions chosen here are different from the ones presented? Please, describe how do you think these choices may affect the results and bias the final distributions obtained.
* The calibration error is all the information provided to quantify the quality of the predictive intervals for regression. It would be helpful to include some other quantification of this, such as the usage of CRPS or other strictly proper scoring rules. In the same line, using metrics such as the Brier score for classification tasks would help getting a more complete picture.
* How does the complexity of the BNNs employed affect the final results? Does using more layers and/or more units improve the results? There have been recent works (e.g. Functional Variational BNNs) regarding artifacts that arise when using large BNNs which make the system not able to properly learn the data. Could these problems be solved as well with this approach since the final prior is constructed using the data?

###### Other comments:

* As an optional suggestion, the article is well written but can be difficult to comprehend at some points. To that end, I would suggest trying to provide qualitative descriptions of some of the quantities in the expressions that later prove to be of importance. As an example, providing some intuition on $\psi(\sqrt{m})$ would help to understand the expression (1). In general, I think the article would benefit from extending a bit the explanations of some parts, especially section 4.
* Typo: section 5, paragraph 2 - "categorigal"
* To have a clearer explanation of figure 1 I would try to include the description of the sinusoidal functions that is already present in the appendix section C.1, since it seems relevant to the text in section 6.1 as well.

---

> ### Author Response · Authors · 2020-11-22
> **Response to AnonReviewer1**
>
> Dear reviewer,
>
> Thank you for your thorough review and for the encouraging feedback.
>
> Since MLAP relies on closed form KL-divergences and we want to keep the benchmark comparison fair, we have restricted our experiments to Gaussian priors with diagonal covariance matrix and a spherical, zero-centered Gaussian hyper-prior. This is the setup that was used in the MLAP paper (Amit & Meir, 2018). However, we believe that having more expressive priors that account for the interactions between the NN weights would be greatly beneficial, not only for the accuracy, but also for the quality of the uncertainty estimates. To this end, Matrix Variate Gaussian distributions (Louizos & Welling, 2016) or Normalizing flows (Rezende & Mohamed, 2015) can be considered. In addition, a less simplistic hyper-prior, such as the horse-shoe prior could be employed to enable a data driven model selection (Ghosh & Doshi-Velez 2017). Compared to MLAP, our proposed method offers the flexibility and computational efficiency to enable the use of more expressive priors and hyper-priors. This is why, in conclusion of the paper, we suggest exploring such avenues in future work.
>
> For the regression experiments we have used a NN with 4 hidden layers of size 32. Since we run experiments with 8 different meta-learning environments and include various baseline methods, the computational burden is very high. To not further increase the computational burden, we haven’t experimented with different model sizes.
>
> However, the fact that we can successfully deploy PACOH-NN on a much larger CNN for the classification tasks seems promising. So far we have not experienced any issues, i.e. that the system is not able to properly learn the data. As we argue in the introduction, a key issue with BNNs is the arbitrary choice of the prior, i.e., zero-centered, spherical Gaussian which may lead to such problems/artifacts. By acquiring the prior in a principled data-driven way, we can most likely avoid such issues with the prior. Nonetheless, we do see the possible adaptation/extension of our framework to function space priors and functional posterior inference as an interesting direction.
>
> Thanks for catching the typo. A reference that provides more details about the set-up in Figure 1 has been included. Also, we have added qualitative descriptions to e.g. $\phi(\sqrt{m})$ to make the writing more accessible.
>
> ---
>
> Amit, Ron, and Ron Meir. "Meta-learning by adjusting priors based on extended PAC-Bayes theory." In: ICML, pp. 205-214. PMLR, 2018.
>
> Danilo Jimenez Rezende and Shakir Mohamed. “Variational inference with normalizing flows”. In: ICML. 2016.
>
> Christos Louizos and Max Welling. “Structured and efficient variational deep learning with matrix gaussian posteriors”. In: ICML. 2016, pp. 1708–1716.
>
> Soumya Ghosh and Finale Doshi-Velez. “Model selection in Bayesian neural networks via horseshoe priors”. In: arXiv preprint arXiv:1705.10388 (2017).

---

### Official Review · AnonReviewer3 · 2020-10-28
**Principled approach to meta-learning with strong empirical results**

**Rating:** 7
**Confidence:** 4

**Review:**

The paper proposes a method for learning BNN priors based on optimising a PAC-Bayes bound for meta-learning, which they call PACOH-NN. This extends previous work on optimising PAC-Bayes bounds (PACOH), and is attractive for being principled in construction and effective in practice. The main claims are that fewer tasks are needed to achieve good performance in terms of both utility and calibration, and that the algorithm comes with performance guarantees.

In the empirical analysis, they firstly show that the method is competitive in terms of utility (for both classification and regression) with baseline methods, but with superior calibration properties. Meanwhile the method is also superior in terms of computational scaling. They also demonstrate the effectiveness on a bandit learning task for vaccine development. Along with proofs, extensive details are given for the experimental setup and training algorithms.


Strengths:
- Principled approach to meta-learning for BNNs
- Efficient algorithm with good scaling properties

Weaknesses:
- A little hard to read in places
- Performance guarantees never shown

Specific comments:
- One of the main motivations for the paper is that the method comes with performance guarantees. E.g. in the related work section
- The use of Hoeffding’s lemma could be very loose, particularly in the cases where values close to 0 or 1 are unlikely
- In the use of the Hoeffding’s lemma, it looks like there should be an extra $m^2$ term in the numerator. Where has this term gone? Similarly, I would expect a factor of $m$ for the sub-Gamma bound in the numerator. This would carry over to the derivations for Thm2 as well, and would also affect the convergence properties
- It seems that you have hyper-priors for the means of the BNN priors only. Are the variances always set to a default value for new tasks? Did you consider putting a hyperprior on the variances/precisions as well? This would feel more in keeping with the Bayesian flavour, and might give the ability to selectively “freeze” (softly) any weights that are stable across all tasks.
- In Figure 1, with only a single training example, I don’t understand how PACOH-NN is able to transfer the knowledge of sinusoidal functions. Since the transfer is only at the hyperprior level, it seems to me that the only possibility is that the tasks are similar enough that the hyperpriors are actually capturing the entire distribution over tasks, so that there is almost no learning required at all on the target task. Can you comment on this?
- You cite the Bayesian MAML paper by Yin et al. However, in the experiments, you only use the regular MAML and first order version, both of which give point estimates. A comparison to BMAML seems to be essential here
- The experiment in 6.2 is a nice addition, and the results are quite impressive on first glance. However, since none of the baselines are meta-learners, we can’t be sure here if the difference is due to the effect of transfer learning or the superior uncertainty quantification. Another meta-learner should be included. There are several meta-bandit algorithms that are available.
- Figure S1 in the appendix is really helpful! If possible, I would prefer to see this in the main paper


Minor comments/Typos:
- P1 variation -> variational
- P2 in the BNN description, the regression model have a noise parameter but the classification model does not. Wouldn’t that imply that the classes would have to be separable (i.e. no mislabelled examples)?
- P3 $\mathcal{H}$ is undefined on first use. Also, in the description the prior is defined as $P \in \mathcal{M}(\mathcal{H})$, but in (1) you have $P \in \mathcal{\Theta}(\mathcal{H})$ and $\mathbb{E}_{\theta \sim P}$. On page 4, you then have $P \in \mathcal{M}(\mathcal{\Theta})$
- Where does Theorem 1 appear in Alquier et al 2016? Is this the correct reference?
- For the sub-Gamma case, you reference Boucheron et al , which is a book. Please make the reference more specific
- Tasks are defined as $\tau_i := (\mathcal{D}_i, S_i)$, but it would seem more accurate to define them as $\tau_i := (\mathcal{D}_i, m_i)$. Not that later you have an expectation where you draw $(\mathcal{D}, m) \sim \tau$ which is also consistent with the latter
- In (7) you use LSE - mention this is LogSumExp
- P8 you say “nearly constant memory and compute”. I’m not sure what “nearly constant” means - is it really linear, but with better scaling constants? Or is it truly constant?
- P12 $p$ undefined in lemma 1
- P18 (53) second line should be $\theta_l$ rather than $\theta_L$

---

> ### Comment · AnonReviewer3 · 2020-11-18
> **Didn't finish a sentence**
>
> Above was:
> "One of the main motivations for the paper is that the method comes with performance guarantees. E.g. in the related work section ..."
>
> This should say:
>
> One of the main motivations for the paper is that the method comes with performance guarantees. E.g. in the related work section other methods criticised as they "fall short of performance guarantees". How strong are the guarantees in practice? Generalisation error bounds tend to be very loose (often vacuous), so while they're a "nice to have" they often don't provide any practical guarantees.

---

> ### Author Response · Authors · 2020-11-22
> **Response to AnonReviewer3**
>
> Dear reviewer,
>
> Thanks a lot for taking the time to write a thorough review and giving detailed tips for improvement. In the following, we try to answer your questions and respond to your comments:
>
> * How tight the bounds and the corresponding performance guarantees depends strongly on setting. For instance in a Bayesian linear regression similar to the toy setup in Section 6 of Catoni et al. (2016), we expect the bounds to be quite tight and useful. However, for large over-parametrized models such as neural networks, as all other PAC bounds for NNs, we do not expect the bounds to be particularly tight. Why neural networks generalize despite their over-capacity is an open problem which our paper does not resolve.
> In general, we see the PAC-Bayesian meta-learning bounds mainly as a guidance for developing a principled algorithm that works well in practice, rather than an artefact that is useful in itself for real-world problems.
>
> * Hoeffding’s lemma: We have checked Case I in Appendix A1 and could not find any mistakes. Note that in equation 18, we have a product with $n$ and $n m$ factors respectively which cancels out with the $n^2$ and $(nm)^2$ in the denominator of the exponent which we get from Hoeffding's lemma and the sub-gamma assumption respectively. This leaves us with $\frac{\beta}{8n}$ and $\frac{\lambda}{nm}$. Does this resolve your question?
>
> * Hyper-prior: We use a hyper-prior over $\phi:= (\mu_P, \log \sigma_P)$, i.e., both the prior mean and log variances (see Section 5.2.). As suggested by you, this allows to selectively “freeze” any weights that can be stable across all tasks.
>
>  * Sinusoids in Figure 1: The details for the sinusoid environment can be found in Appendix C.1.1. A visual representation is also available in Figure S2a. Indeed, the tasks do share a lot of similarities which makes meta-learning possible, i.e. they all slope upward and have a sinusoidal shape. However, the tasks differ in four parameters (slope, amplitude and translation in x/y direction) and contain an additive gaussian observation noise. This makes task identification non-trivial and thus reasoning about epistemic uncertainty crucial.
>
> * BMAML: Upon your request, we have implemented Bayesian MAML and are currently running the experiments using it – The results for BMAML on the regression tasks have already been incorporated into the revised version of the paper. The BMAML experiments on the classification tasks are still running. We will add the corresponding results for the paper by Tuesday, Nov 24.
>
> * In fact, to ensure a fair comparison, for GP-UCB and GP-TS, we meta-learn the GP prior parameters by minimizing the marginal log-likelihood on the five meta-training tasks (see Appendix C4). Did you have any specific meta-bandit algorithm in mind that goes beyond that?
>
>
> Minor comments:
>
> * The classification (labelling) noise is implicit in the softmax-categorical class probabilities which are determined by the logits which the neural network outputs. Technically, one could add a temperature/scaling parameter to the softmax/logits. However, the neural network can scale the logits itself - thus no need for an extra parameter. If the logits values are small in magnitude, the categorical distribution has higher entropy corresponding to more labelling noise / aleatoric uncertainty. If the logit values are large in magnitude the softmax tends towards a max operation and thus little labelling noise.
>
> * Regarding Theorem 1, the reference should be correct. Theorem 1 in our paper corresponds to Theorem 4.1 in Alquier et al 2016.
>
> * We have added the chapter and pages to the Bucheron et al. reference.
>
> * Thank you for noticing the ambiguity regarding the complexity analysis. Since the favorable scaling properties are a key advantage of PACOH-NN over MLAP, we have added a memory and computation complexity analysis to Section 5 of the revised paper. In the mini-batched version of the algorithm, both the memory and computational complexity are constant w.r.t. the number of tasks n.
>
> * Thanks for catching the typos and the undefined abbreviations/variables – they have been fixed.
>
> Thanks again for your constructive feedback. We hope that our corresponding changes have improved the paper.
>
> ---
>
> Germain, Pascal, Francis Bach, Alexandre Lacoste, and Simon Lacoste-Julien. "PAC-Bayesian theory meets Bayesian inference." In NIPS, pp. 1884-1892. 2016.

---

### Official Review · AnonReviewer4 · 2020-10-31
**Paper is well-written but its contribution is marginal**

**Rating:** 6
**Confidence:** 4

**Review:**

The paper addresses the problem of learning data-driven priors for Bayesian neural networks. Assuming zero-centered Gaussian priors for BNNs often results in poor generalization and uncertainty quantification, whereas choosing informative priors is challenging due to limited interpretability of network weights. To be able to overcome these issues, the authors propose a meta-learning framework based on PAC-Bayesian theory, in which they optimize a PAC bound called PACOH in the space of possible posterior distributions of BNN weights. Unlike the previous approaches in the literature, their approach doesn’t rely on nested optimization schemes, instead they directly minimize PAC bound via a variational algorithm called PACOH-NN which is based on SVGD and reparameterization trick.

The main paper and its comprehensive appendix are well written mostly, especially the PAC-Bayes review is very concise and elaborate for a conference paper. After the introduction of related concepts about PAC-Bayes bounds and meta-learning, their application to the derivation of PACOH bound for finding a hyper-posterior expressed so naturally. Proposed methodology sounds, and claims of the authors seem correct. Experimental results both on synthetic and real-world data seem fine, and there seems to be no fallacy in empirical evaluations. Also, experimental methodology is discussed thoroughly both in the main paper and appendix. Empirical results from various experimental setups are sufficient to show that their framework improves both predictive accuracy and the uncertainty estimates compared to several popular meta-learning approaches with additional computational advantages.

As a researcher with superficial knowledge of the field, I didn't have a major difficulty in following the paper, despite the use of heavy mathematics in methodology. Mathematical notation is clear and consistent (except the usage of $M(\Theta)$ and $M(H)$ interchangeably). Derivations in both the main paper and the appendix are easy to follow. Also, relation to prior work is clearly addressed, relative advantages and disadvantages of the proposed methodology are discussed.

In my opinion, the problem of meta-learning informative priors for Bayesian neural networks will certainly be of interest to the ICLR community. On the other hand, paper’s contribution to the work of Rothfuss et al. seems limited. Authors state that the main contribution of the paper is their practical variational algorithm for PACOH bound. In order to tractably optimize PACOH for a BNN, they employ a Stein gradient variational descent algorithm by utilizing reparameterization trick. However, the specifics of their algorithm are presented very quickly, which makes it difficult to grasp the actual novelty of PACOH-NN. For instance, neither the advantages of this particular optimization scheme over the previous methods, nor the theoretical guarantees regarding convergence and approximation quality for PACOH-NN aren’t discussed.

---

> ### Author Response · Authors · 2020-11-22
> **Response to AnonReviewer4**
>
> Dear reviewer,
>
> Thank you for your review – we are happy to hear that you enjoyed reading our paper and consider it of interest to the ICLR community.
>
> While our approach indeed builds on the PAC-Bayesian meta-learning methodology developed in Rothfuss et al. (2020), the present submission goes substantially beyond it, as elaborated below.  Crucially, in terms of algorithms, previous work was restricted to tractable base learners (such as Gaussian processes).  We theoretically show how to lift these restrictions, and present extensive empirical demonstrations on (intractable) Bayesian neural network base learners.
>
> We agree though that in the submitted version of the paper, the PACOH-NN algorithm is presented very quickly and discussed insufficiently. In order to address these shortcomings, we have significantly extended Section 5 and its counterpart in Appendix B. In particular, we have added:
>
>
>
> *   a detailed discussion of the properties / approximation quality of our proposed generalized marginal log-likelihood estimator
> *   a description of the proposed PACOH-NN algorithm
> *   a memory and computation complexity analysis thereof
> *   a discussion of PACOH-NN’s advantages over previous methods
>
> to the revised version of the paper.
>
> Moreover, we want to emphasize that, thanks to our improved proof methodology, our proposed  bounds are tighter than the ones in Rothfuss et al. (2020). We extended the corresponding discussion under Theorem 2 in the reviser paper.
>
> Overall, the key algorithmic contribution of our paper over Rothfuss et al. (2020) is the numerically stable score estimator for the generalized marginal log-likelihood and our analysis which shows that despite the approximation error, we still minimize a valid upper bound on the transfer error. This makes PAC-Bayesian meta-learning both tractable and scalable for a much larger array of models, including BNNs. In contrast, Rothfuss et al. is restricted to models with closed-form marginal log-likelihood which severely restricts its use. In contrast, PACOH-NN can also be used for tasks such as image classification or high-dimensional Bayesian optimization (e.g. the experiment in Section 6.2.) to which the method of Rothfuss et al. (2020) is not applicable.
>
> Thus, we consider PACOH-NN as an algorithm of high practical relevance which we try to showcase in our extensive experiments on real-world meta-learning problems.
>
> Finally, we want to thank the reviewer for the provided feedback, which allowed us to provide a better emphasis on the strengths and novelty of our approach.

---

### Author Response · Authors · 2020-11-22
**Changes in the revised version of the paper**

Addressing the reviewers feedback, we have made the changes to the paper and ran further experiments. In summary, the most significant changes were the following:



*   In Section 4, we have extended the discussion below Theorem 2, detailing how our bound improves upon previous bounds for meta-learning.
*   In Section 5 we have added :
    *   a detailed discussion of the properties / approximation quality of our proposed generalized marginal log-likelihood estimator
    *   a description of the proposed PACOH-NN algorithm
    *   a memory and computation complexity analysis thereof
    *   a discussion of PACOH-NN’s advantages over previous methods
*   Section 6: Upon the reviewers request, we ran experiments with BMAML and have included it into our empirical benchmark study. Corresponding results on our regression benchmark tasks have been added to Table 1 & 2. The experiment results for the classification tasks will be added soon to Table 3.
*   Appendix:
    *   We have added a mini-batched version of the PACOH-NN algorithm to Appendix B.1.
    *   We have extended Appendix B.3, deriving and discussing the statistical properties of the general marginal log-likelihood estimator.
    *   We have added Section C.4. which provides further details on the bandit experiment in Section 6.2.

---

> ### Author Response · Authors · 2020-11-23
> **BMAML experiment results for classification  have been added**
>
> The experiment results BMAML on the classification tasks have been added to Table 3 in the revised version of the paper.

---

> ### Author Response · Authors · 2020-11-25
> **Experiment results for GP-based method of Rothfuss et al. (2020) have been added to revised paper**
>
> In response to the suggestion of AnonReviewer2, we conducted experiments with the GP-based method of Rothfuss et al. (2020) on the five regression environments. The respective results have been added to Table 1 & 2 and the experiment description / discussion has been modified accordingly.

---

### Decision · Program_Chairs · 2021-01-07
**Final Decision**

**Decision:**

Reject

**Comment:**

The paper addresses the problem of prior selection in Bayesian neural networks by proposing a meta-learning framework based on PAC-Bayesian theory. The authors optimize a PAC bound called PACOH in the space of possible posterior distributions of BNN weights. The method does not rely on nested optimization schemes, instead, they directly minimize PAC bound via a variational algorithm called PACOH-NN which is based on SVGD and the reparameterization trick.  The method is evaluated on experiments with both synthetic and real-world data showing improvements in both predictive accuracy and uncertainty estimates.

Initially many reviewers were positive about the paper. However, it was noticed by one reviewer that the submitted paper presents a very significant overlap with

Jonas Rothfuss, Vincent Fortuin, and Andreas Krause. PACOH: Bayes-Optimal Meta-Learning with PAC-Guarantees. arXiv, 2020.

Another reviewer mentioned that they were actually reviewing for AISTATS the above manuscript by Rothfuss et al. The ICLR program chairs were contacted for a possible violation of the dual submission policy for ICLR:

"Submissions that are identical (or substantially similar) to versions that have been previously published, or accepted for publication, or that have been submitted in parallel to this or other conferences or journals, are not allowed and violate our dual submission policy."

The ICLR program chairs decided that the similarities between the two papers are not enough to issue a desk-rejection. However, in the discussion period,  three reviewers out of 4 pointed out that, even though the authors did revise Sections 4 and 5 in the current version, these modifications do not seem to be strong enough to make up for the really strong overlaps between the two papers. The reviewers agreed on rejection and stated that this paper should either be merged with the Rothfuss et. al. one (assuming the authors are the same), or its content should be developed to the point of making both of them clearly distinct.